# Layers at Similar Depths Generate Similar Activations Across LLM Architectures

**Christopher Wolfram**
Department of Computer Science
University of Chicago
Chicago, IL 60637, USA
chriswolfram@uchicago.edu

**Aaron Schein**
Department of Statistics
University of Chicago
Chicago, IL 60637, USA
schein@uchicago.edu

## Abstract

How do the latent spaces used by independently-trained LLMs relate to one another? We study the nearest neighbor relationships induced by activations at different layers of 24 open-weight LLMs, and find that they 1) tend to vary from layer to layer within a model, and 2) are approximately shared between corresponding layers of different models. Claim 2 shows that these nearest neighbor relationships are not arbitrary, as they are shared across models, but Claim 1 shows that they are not "obvious" either, as there is no single set of nearest neighbor relationships that is universally shared. Together, these suggest that LLMs generate a progression of activation geometries from layer to layer, but that this entire progression is largely shared between models, stretched and squeezed to fit into different architectures.[1]

## 1 Introduction

Instead of trying to understand the internal mechanisms of specific language models, we study structural similarities in the activations generated by models of different architectures. In particular, how do the latent spaces used by different models relate to one another, and do they have any universal properties?

Consider the following experiment. Let $\mathcal{D}$ be a set of prompts. We pass each prompt in $\mathcal{D}$ to a language model $M$ and collect the activations generated at a particular layer[2] $\ell$. For any prompt $t \in \mathcal{D}$, we can then ask: which are the $k = 10$ other prompts that yielded activations most similar to this one? This gives a set of the $k$ nearest neighbors of $t$ among the other elements of $\mathcal{D}$. Repeating this for all prompts in $\mathcal{D}$ gives a collection of nearest neighbor relationships that encode which prompts are nearest to which others according to layer $\ell$ in the model $M$.

By comparing nearest neighbor relationships across a wide range of models, we make two findings that seem unremarkable on their own, but together are quite surprising:

**Claim 1.** Activations collected at different depths within the same model tend to have **different** nearest neighbor relationships.

**Claim 2.** Activations collected at corresponding depths of *different* models tend to have **similar** nearest neighbor relationships.

Claim 2 shows that nearest neighbor relationships are not arbitrary, as they are shared across independently-trained models. Claim 1, however, shows that they are not "obvious" either, as nearest neighbor relationships vary with depth. Put another way, Claim 1 shows that nearest neighbor relationships are sensitive to differences between layers within a single model, while Claim 2 shows that activations at corresponding depths of different models are similar.

---

[1]Code available at https://github.com/chriswolfram/unireps

[2]We use the term "layer" to refer to an entire decoder module. See Section 3.2 for details on how we extract activations from transformer models.

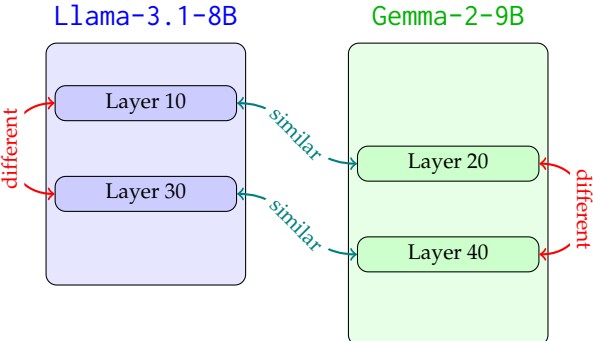

Figure 1: Layers at different depths generate activations with different nearest neighbor relationships (Claim 1), but layers at corresponding depths of different models generate activations with similar nearest neighbor relationships (Claim 2). The particular example used in Sections 1 and 2 is shown here.

Such nearest neighbor relationships reflect the local geometry of the embeddings generated at a particular layer, and are closely related to operations used in transformer models. Together, Claims 1 and 2 suggest that transformer-based language models use a progression of distinct local geometries from layer to layer, and that this entire progression of local geometries is (largely) shared across models regardless of architecture. In other words, independently-trained language models tend to go through a similar sequence of intermediate representations when given the same prompt.

To give a concrete example, let $t$ be this text from OpenWebText (Gokaslan et al., 2019), a dataset mostly containing internet news articles:

`Training For Ice and Mixed Climbing Series brought to you by Furnace Industries continues...`

According to layer 10 of `Llama-3.1-8B`, these are the 3 nearest neighbors to that text $t$ among a sample of 2047 other texts from OpenWebText:

- `There have been a lot of crazy reactions from the democrats now that Donald Trum...`
- `Prosecutor Christina Weilander filed the charges with Södertörn district court o...`
- `7 questions on the Great British Bake Off Great British Bake Off Quiz Then there...`

At first glance, these nearest neighbors look arbitrary, and it is not clear what relationship they have with the original input or with one another. That makes it all the more surprising that layer 20 of `Gemma-2-9B` gives exactly the same 3 nearest neighbors (in line with Claim 2). Moreover, layer 30 of `Llama-3.1-8B` gives another completely different set of nearest neighbors (in line with Claim 1), but many of them are the same as the nearest neighbors given by layer 40 of `Gemma-2-9B` (Figure 1).

`Llama-3.1-8B` and `Gemma-2-9B` are architecturally different models independently trained by different companies, so it is surprising that we not only find similarities in their latent representations, but that these similarities only appear at corresponding depths. (See Section 2 for more investigation of this specific example.)

We study this phenomenon systematically by generating the $n_1 \times n_2$ *affinity matrix*, which stores the similarity of nearest neighbor relationships implied by each pair of layers $(i, j)$ with layer $i \in [n_1]$ from model $M_1$ and layer $j \in [n_2]$ from model $M_2$. The resulting affinity matrices exhibit diagonal structure, with higher similarity among layers at corresponding depths (implying Claim 2), and lower similarity among pairs of layers from different depths (implying Claim 1).

This paper studies the emergence of diagonal structure in affinity matrices across 24 open-weight language models, with sizes spanning from 1B to 70B parameters. We find that diagonal structure is present when comparing a wide range of models (Section 4), and that this diagonal structure is statistically significant for every pair of models we consider (Section 4.1).

## 1.1 Related work

There is an extensive literature on representational similarity in neural networks (Kornblith et al., 2019; Klabunde et al., 2024), and with rising interest in universality in LLMs (Olah et al., 2020; Rai et al., 2025) more recent work has begun to apply representational similarity methods to large transformer models (Huh et al., 2024; Klabunde et al., 2023). However, none have systematically studied affinity matrices comparing layers of architecturally differ­ent LLMs or the resulting diagonal structure. Most either 1) compared layers within a single model (Vulić et al., 2020), 2) compared layers between two copies of a model trained with different random seeds (Nguyen et al., 2022; Gurnee et al., 2024), 3) focused on CNNs or very small language models (Raghu et al., 2017), 4) used a representational similarity measure that did not reveal diagonal structure (Lan et al., 2024; Baek et al., 2024; Brown et al., 2023), or some combination thereof (Kornblith et al., 2019; Nguyen et al., 2021). Of particular note is the work of Wang et al. (2025), who used sparse autoencoders to study representational sim­ilarity of Pythia-160M and Mamba-130M and noted that layers at similar depths tended to be more similar to one another. Also notable is the work of Wu et al. (2020), who measured the representational similarity of layers from a range of early LLMs. They found that early layers tended to be more similar between models than later layers, and observed diagonal structure in some model comparisons (notably in a comparison of BERT-base and BERT-large).

## 2 Motivating experiment

Let $\mathcal{D}$ be a random sample of 2048 texts from OpenWebText (Gokaslan et al., 2019), a dataset of internet text dominated by news articles. As before, let $t$ be the following text from OpenWebText (which has been truncated for space):

```
Training For Ice and Mixed Climbing Series brought to you by Furnace Industries continues...
```

We feed all of the texts in $\mathcal{D}$ to Llama-3.1-8B and collect the activations generated at layer 10. We then compute the 10 elements of $\mathcal{D}$ that generated activations most similar (by cosine distance) to those generated when $t$ was inputted. In other words, we compute the 10 nearest neighbors of $t$ according to layer 10 of Llama-3.1-8B.

We then repeat this process for layer 20 of Gemma-2-9B, and make a Venn diagram showing which elements of $\mathcal{D}$ were considered to be nearest neighbors to $t$ according to the two layers (Figure 2).

Llama-3.1-8B (layer 10)

Gemma-2-9B (layer 20)

Figure 2: Venn diagram showing the 10 nearest neighbors to the text $t$ among the 2048 elements of $\mathcal{D}$ according to layer 10 of Llama-3.1-8B (in the blue box) and layer 20 of Gemma-2-9B (in the green box). Entries that both layers agreed were nearest neighbors are marked with ✅, while others are marked with ❌.

In this example, these two layers from completely different models agreed on 7 out of the 10 nearest neighbors. This is all the more surprising because, at first glance, these texts seem to have little to do with the text $t$ to which they are all nearest neighbors, nor do they share much with one another. They appear arbitrary, but are evidently not arbitrary because they are independently identified by different layers of different models. On closer inspection, these texts do share a common feature in that many end with a variant of "Follow _ on Twitter." It could be that the activations generated at layer 10 of `Llama-3.1-8B` and layer 20 of `Gemma-2-9B` are both dominated by mentions of Twitter.

The previous example could give the impression that nearest neighbor relations in activations are dominated by superficial but obvious features, and that agreement between models is a trivial consequence of both models identifying the same obvious features. However, this idea is dispelled by looking at different layers.

We now compute the nearest neighbors to $t$ according to layer 30 of `Llama-3.1-8B` and layer 40 of `Gemma-2-9B` (Figure 3).

### Llama-3.1-8B (layer 30)

```
❌ Long Forgotten London Old House on Tower Hill There is the London we know and...
❌ Looking for news you can trust? Subscribe to our free newsletters. Texas alre...
❌ Sure, I love tea. And I admire those who go caffeine-free. But I have become ...
❌ Wrightwood. Cal. 21 October, 1949 Dear Mr. Orwell, It was very kind of you to...
✅ About DONALD TRUMP and HILLARY CLINTON, Skewered by Their Own Words with Funn...
✅ In a sign of the Queen's determination to master new technology, Buckingham P...
✅ In case you were living under a rock, today Manchester City announced the sig...
✅ Low-Riders at the Museum El Paso Museum of Art Saturday, May 10, 2014 from 10...
✅ Mitt Romney's hurdle in winning the love/respect/admiration/fear of his party...
✅ We all love a selfie, regardless of the occasion. But where are the most popu...
❌ On Monday, November 25, UrtheCast will present live the launch of its two cam...
❌ Review: "Sight Unseen" @ NJT The art world is full of beautifully complex, cr...
❌ Since launching The Athletic NHL and The Athletic Toronto more than a year ag...
❌ window._taboola = window._taboola || []; _taboola.push({ mode: 'thumbnails-c'...
```

Gemma-2-9B (layer 40)

Figure 3: Venn diagram showing the 10 nearest neighbors to the text $t$ according to layer 30 of `Llama-3.1-8B` (in the blue box) and layer 40 of `Gemma-2-9B` (in the green box). The two layers agree on 6 out of the top 10 nearest neighbors.

Notably, the set of nearest neighbors according to layer 30 of `Llama-3.1-8B` is completely disjoint from the set of nearest neighbors identified by layer 10 of the same model. If there was some obvious feature of $t$ that was identified by the earlier layer (such as mentions of Twitter), it is not being identified by this later layer of the same model. In addition, this new list of nearest neighbors is not arbitrary either, as it shares most of its elements with the nearest neighbors identified by later layers of `Gemma-2-9B`.

That the nearest neighbors according to layer 10 of `Llama-3.1-8B` and layer 20 of `Gemma-2-9B` are so similar (as well as layer 30 of `Llama-3.1-8B` and layer 40 of `Gemma-2-9B`) is in line with Claim 2 (that activations from corresponding layers of different models induce similar nearest neighbor relationships). In contrast, that the nearest neighbors according to layer 10 and layer 30 of `Llama-3.1-8B` are so different is in line with Claim 1 (Figure 1).

For layer 10 of `Llama-3.1-8B` and layer 20 of `Gemma-2-9B`, we identified a superficial property of the inputs that could be dominating the geometry of the latent spaces (i.e., that they mention Twitter). We found no such superficial feature relating nearest neighbors according to layer 30 of `Llama-3.1-8B` and layer 40 of `Gemma-2-9B`, but it is possible that there is one. However, neither of these findings would explain away the main observation: Why should independently-trained models identify the same (perhaps superficial) properties to focus on, especially if different layers in the same models focus on different properties?

We do not purport to know why the activations at these layers of these models induced these particular nearest neighbors, only that some of these sets are remarkably similar and others are disjoint. These patterns of similarity and dissimilarity are the object of our study.

## 3 Representational similarity

In order to study this systematically, we first define some notation for embeddings, nearest neighbors, and representational similarity.

**Definition 1 (Embedding functions)** *An* embedding function *is a map from an input space $\mathcal{X}$ to $\mathbb{R}^d$ for some embedding dimension d.*

The embedding functions we use extract activations from transformer models.

**Definition 2 (Representational similarity measures)** *Let $f : \mathcal{X} \to \mathbb{R}^{d_1}$ and $g : \mathcal{X} \to \mathbb{R}^{d_2}$ be embedding functions. A* representational similarity measure *is a function*

$$s : (\mathcal{X} \to \mathbb{R}^{d_1}) \times (\mathcal{X} \to \mathbb{R}^{d_2}) \to \mathbb{R}$$

*that maps a pair of embedding functions to $\mathbb{R}$.*

Note that we do not restrict ourselves to cases where $d_1$ and $d_2$ are equal, as models of different architectures often have different embedding dimensions.

We focus primarily on measuring similarity between nearest neighbor relationships among activations, but our framework could be applied to any measure of representational similarity satisfying Definition 2, such as CKA (Kornblith et al., 2019) or one of many others (Klabunde et al., 2024).

The central problem of measuring representational similarity is that neural networks can generate activations that differ in trivial ways. For example, one neural network might generate activations identical to another, except that the axes are permuted, or the signs are flipped, etc. The challenge of measuring representational similarity is in choosing what properties of representations are arbitrary (e.g., the sign or order of dimensions), and what properties are fundamental (e.g., the nearest neighbor relationships).

In this work, we study *patterns* of similarity: that layers at corresponding depths in different models tend to generate activations that share properties that layers at non-corresponding depths do not. As such, the particular choice of representational similarity measure is not our focus. We do not claim that the nearest neighbor-based measure we use is the "correct" way to measure representational similarity (whatever that would mean), but that it shows that there *exist* properties of activations that are shared according to these patterns.

### 3.1 Nearest neighbor relationships

We measure the extent to which nearest neighbor sets agree using the mutual *k*-nearest neighbors (mutual *k*-NN) representational similarity measure described by Huh et al. (2024).

**Definition 3 (*k*-nearest neighbors)** *Let $\mathcal{U}_k^{(\mathcal{D})}$ be the k-nearest neighbor function*

$$\mathcal{U}_k^{(\mathcal{D})}(x, f) \triangleq \underset{\mathcal{U} : \mathcal{U} \subseteq \mathcal{D} \setminus \{x\}, \|\mathcal{U}\| = k}{\operatorname{argmin}} \sum_{x' \in \mathcal{U}} d\left(f(x), f(x')\right) \tag{1}$$

*for a distance function d. Unless otherwise stated, we use the cosine distance.*

Intuitively, $\mathcal{U}_k^{(\mathcal{D})}(x, f)$ gives the $k$ elements of a dataset $\mathcal{D}$ that are closest to $x$, after they have been mapped to vectors by $f$.

**Definition 4 (Mutual $k$-nearest neighbors)** *The* mutual $k$-nearest neighbors *representational similarity measure $\mathcal{M}_k^{(\mathcal{D})}$ is defined by*

$$\mathcal{M}_k^{(\mathcal{D})}(f,g) \triangleq \frac{1}{k\|\mathcal{D}\|} \sum_{x\in\mathcal{D}} \left\| \mathcal{U}_k^{(\mathcal{D})}(x,f) \cap \mathcal{U}_k^{(\mathcal{D})}(x,g) \right\|$$

*for an input dataset $\mathcal{D}$.*

Mutual $k$-NN measures the similarity of the nearest neighbor relationships induced by $f$ to those induced by $g$. For a set of inputs, it gives the expected fraction of the $k$-nearest neighbors that both embedding functions agree on. (Note that $\mathcal{M}_k^{(\mathcal{D})}$ takes two embedding functions and returns a scalar, and therefore satisfies Definition 2.)

In general, when we refer to layers being "similar" to one another, we mean that they achieve a high mutual $k$-NN similarity score.

## 3.2 Activations in transformer models

Transformer models generate many activations during their operation, and can be probed at several different points. We restrict ourselves to the activations generated at the end of each decoder module (which are sometimes referred to as "layers").

Thus, we can think of a transformer model $M$ with $n$ layers as providing the embedding functions $f_M^{(1)}, f_M^{(2)}, \ldots, f_M^{(n)} : \mathcal{T} \to \mathbb{R}^d$ which map from texts $\mathcal{T}$ to vectors. Each embedding function gives the activations at the end of a layer in $M$ on the last token of the input.

## 3.3 Affinity matrices

**Definition 5** *An* affinity matrix *for models $M_1$ and $M_2$ is a matrix $A$ such that*

$$A_{i,j} = s(f_{M_1}^{(i)}, f_{M_2}^{(j)}) \tag{2}$$

*given some representational similarity measure $s$. Unless otherwise noted, we use $\mathcal{M}_{10}^{(\mathcal{D})}$ for $s$, which measures similarity with mutual $k$-NN with $k = 10$. The dataset $\mathcal{D}$ must also be specified.*

If $M_1$ and $M_2$ have $n_1$ and $n_2$ layers respectively, the resulting matrix $A$ has dimensions $n_1 \times n_2$, with each entry $A_{i,j}$ storing the representational similarity of the layer $i$ of $M_1$ and layer $j$ of $M_2$. Because we use mutual $k$-NN as our representational similarity measure, this means that $A_{i,j}$ stores the similarity of the nearest neighbor relationships generated by those two layers.

It is worth noting that mutual $k$-NN is always measured with respect to a dataset of inputs $\mathcal{D}$. This extends to affinity matrices generated with mutual $k$-NN as well.

We are not the first to generate such matrices (Kornblith et al., 2019; Nguyen et al., 2021; 2022; Gurnee et al., 2024; Raghu et al., 2017; Wang et al., 2025; Abnar et al., 2019; Brown et al., 2023; Wu et al., 2020; Vulić et al., 2020), though they have most often been used to compare layers either from the same model or from multiple instances of the same architecture trained with different random seeds.

## 4 Affinity matrices reveal diagonal structure

In Section 2, we saw a single example where activations collected at different depths within the same model had different nearest neighbor relationships (Claim 1), and activations collected at corresponding depths of different models had similar nearest neighbor relationships (Claim 2). Do these results generalize to other inputs, layers, models, etc.?

The mutual $k$-NN representational similarity measure captures the extent to which two layers generate activations with similar nearest neighbor relationships. By generating

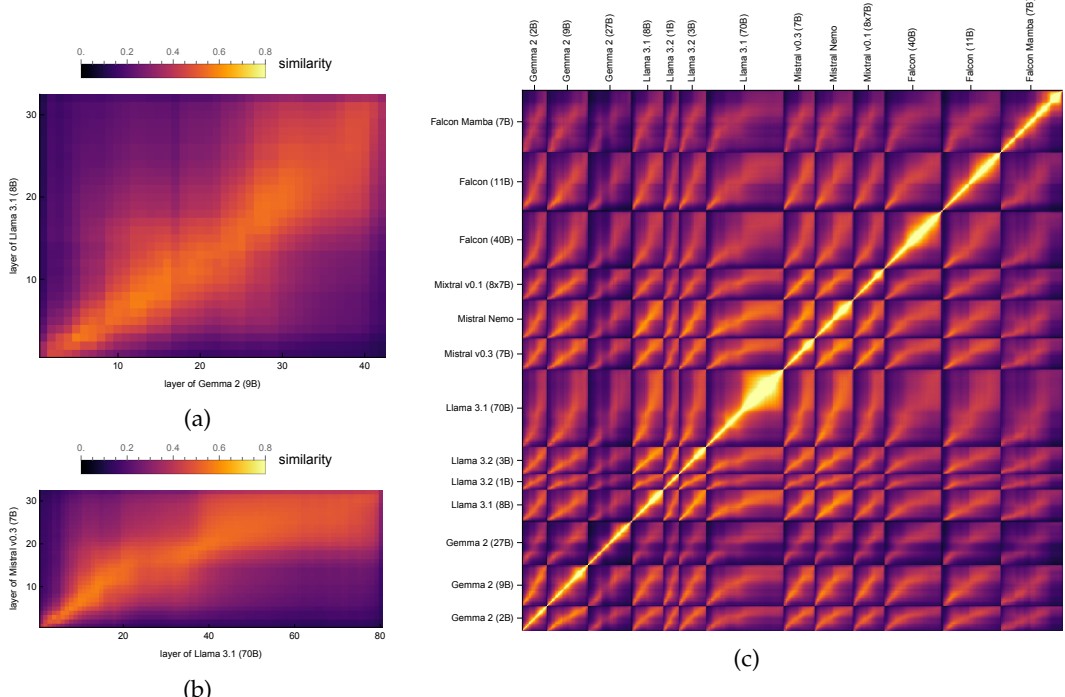

Figure 4: Layers at corresponding depths tend to generate similar activations on internet text, as represented by diagonal structure in affinity matrices. 4a The affinity matrix of `Gemma-2-9B` and `Llama-3.1-8B` on a sample of texts from OpenWebText. Each cell corresponds to a pair of layers, with one from `Gemma-2-9B` and the other from `Llama-3.1-8B`. The value at that cell corresponds to the similarity of nearest neighbor relations induced by the activations at those layers. 4b The same as 4a, but for `Llama-3.1-70B` and `Mistral-7B`. 4c An array of affinity matrices comparing activations on web text across a wide range of models, all of which show diagonal structure. (See Figure 17 for an extended version and Figure 25 for a version that resizes all affinity matrices to squares for comparability.)

affinity matrices that show the mutual *k*-NN similarity of every pair of layers across a range of models, we can see whether and how the observations of Section 2 generalize.

We consider 24 open-weight models developed by Google (Gemma Team et al., 2024), Meta (Grattafiori et al., 2024), Mistral (Jiang et al., 2023), Microsoft (Abdin et al., 2024), and TII (Almazrouei et al., 2023). (See Appendix F for the full list of models.) For every pair of models, we generate an affinity matrix (Definition 5) which shows how similar the nearest neighbor relationships are between every pair of layers in those models. We measure similarity using mutual *k*-NN (Definition 4) with $k = 10$, and using a dataset of 2048 texts randomly sampled from OpenWebText (Gokaslan et al., 2019).

We visualize a selection of the resulting affinity matrices in Figure 4. The most striking feature of these affinity matrices is their *diagonal structure*, which is present across a wide range of model comparisons (Figure 4c). This diagonal structure implies that layers at *proportionally* similar depths generate more similar activations to one another than other pairs of layers, just as seen in Section 2. This indicates that models generate distinct geometries of activations at different depths, but that activations at a given depth are similar across diverse models. Put another way, language models generate a progression of distinct activation geometries (Claim 1), and this entire progression is largely shared between models (Claim 2).

**Models generate similar progressions of embeddings, but stretched or squeezed** The most similar layers between models cannot align one-to-one because the models being compared have different numbers of layers. In the case of `Llama-3.1-8B` and `Gemma-2-9B`

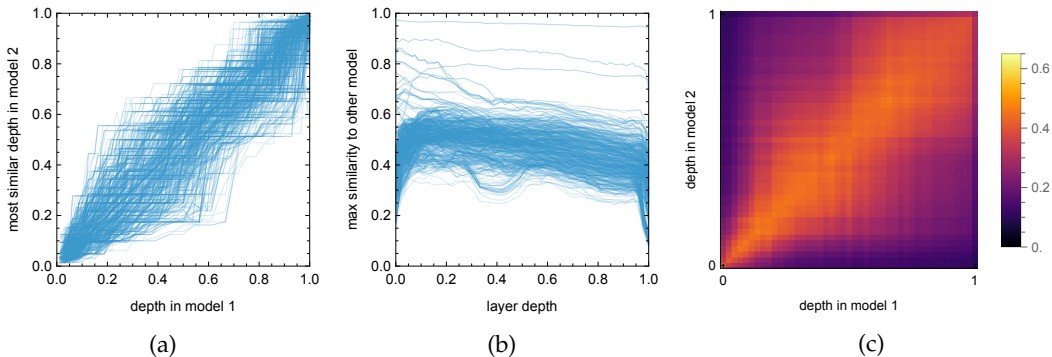

Figure 5: 5a Maximally similar layers appear at proportional depths. Each line represents a pair of models $M_1$ and $M_2$, and for each layer in $M_1$ it shows the depth of the most similar layer in $M_2$. 5b Layers at all depths have similar counterparts in other models. Each line represents a pair of models $M_1$ and $M_2$, and shows how similar the most similar layer of $M_2$ is to each layer of $M_1$. 5c The mean affinity matrix when all affinity matrices have been resized to fit into a square.

(Figure 4a), the most similar layers are not at the same absolute depths (e.g., layer 10 of each model), but instead at similar depths relative to the total number of layers in the model (e.g., 30% of the way through). This pattern persists across a range of model pairs, and the correspondence between maximally similar layers does not deviate far from proportionality in any of the model pairs we consider (Figures 5a and 5c).

**Layer compression**   When comparing models of vastly different sizes, there is yet more flexibility in the correspondence between layers. For example, consider the affinity matrix comparing `Llama-3.1-70B` and `Mistral-7B`, which have 80 and 32 layers respectively (Figure 4b). The early layers appear to correspond almost one-to-one, but the middle layers of `Llama-3.1-70B` appear highly compressed in `Mistral-7B`. The later layers are all relatively similar to one another, and so show less diagonal structure. This general pattern is typical, though often stretched and squeezed to fit into models with different numbers of layers (Figures 4c and 17).

Existing work has argued that layers at different depths in transformer language models perform different functions, for example, with early layers dedicated to parsing the input and resolving basic grammar, and later layers dedicated to finding the next token of the response etc. (Lad et al., 2024; Tenney et al., 2019; Durrani et al., 2020; Kaplan et al., 2025; Cheng et al., 2025). It is possible that these functions are reflected in the correspondence between layers of large and small models. Language models need to be able to parse their input, regardless of size, and so the first 15 to 20 layers generally correspond one-to-one with little compression. Middle layers, however, could be responsible for more abstract and optional work that can be highly compressed in smaller models, as is reflected in the affinity matrices.

**There is similarity at every depth**   Intuitively, one might expect there to be higher similarity between models when comparing the earliest and latest layers, as these are the most closely tied to model behavior and the training process. Indeed, in their experiments, Wu et al. (2020) found that earlier layers were more similar across models than later layers. However, we find that the absolute similarity between layers does not vary much with depth, with middle layers being just as similar to one another as early and late layers (Figure 5b). Even in architecturally identical models trained with identical data (but with different random initialization), there is no intrinsic guarantee that their latent activations be similar, but here we find that similar latent activations are generated at every depth.

**Sensitivity to the experimental setup**   The experiments above all look at activations generated on inputs sampled from OpenWebText. In Appendix G, we generate affinity matrices for a range of input datasets, including IMDB movie reviews (Maas et al., 2011), text

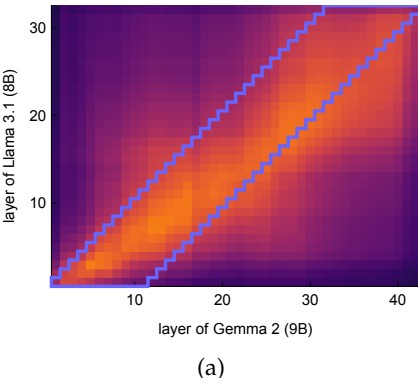
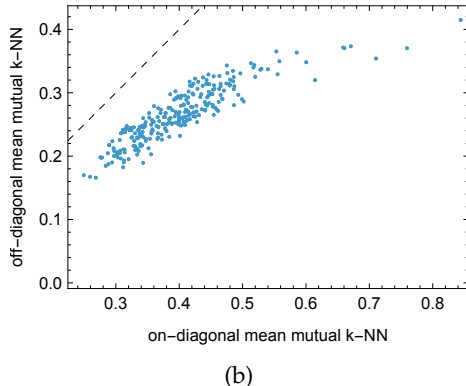

(a)                                              (b)

Figure 6: 6a An affinity matrix (from Figure 4a) overlaid with the generalized diagonal region (red). 6b Layers pairs that lie on the generalized diagonal have higher mean similarity than those that do not. Each point represents a pair of models, with its position determined by the mean similarity of its layer pairs that lie on and off the generalized diagonal.

from books in English and German (Tiedemann, 2012), and various LLM evaluations (Zhou et al., 2023; Hendrycks et al., 2021). The diagonal structure observed with OpenWebText is generally present for other distributions of web text (e.g., IMDB movie reviews). Behavior on other input distributions is investigated in Appendix A.

We also perform experiments with other values of $k$ and find that the choice of $k$ shifts the measured mutual $k$-NN similarity, but does not affect diagonal structure, even in extreme cases like $k = 1$ (Appendix D).

### 4.1 Statistical test of diagonal structure

For each model pair, we perform a test of diagonal structure based on comparing the mean similarity of layer pairs near the diagonal to the mean similarity of those away from the diagonal. We first have to define what we mean by "near" the diagonal.

**Generalized diagonal** For a square matrix $A$, the main diagonal contains those elements $A_{i,j}$ where $i = j$. However, many of the affinity matrices we generate are not square. For a rectangular matrix $A$ with dimensions $n \times m$, we define the *generalized diagonal* to be those elements $A_{i,j}$ satisfying $j \leq i \leq j + n - m$ (where $n \geq m$ without loss of generality). In the case where $A$ is square (i.e., $n = m$), the generalized diagonal is exactly the usual diagonal. An example of the generalized diagonal is shown in Figure 6a.

For each affinity matrix (generated with inputs on OpenWebText), we compute the mean similarity of those pairs of layers that are included in the generalized diagonal, and compare them to the mean similarity of all other pairs (Figure 6b). For every pair of models, we find that the on-diagonal mean similarity is higher than the off-diagonal mean similarity.

**Naive $t$-test** For each model pair $(M_u, M_v)$ we then have the null hypothesis $H_0^{u,v} : \mu_{u,v} \leq \mu'_{u,v}$ where $\mu_{u,v}$ denotes the mean similarity of on-diagonal cells in the affinity matrix comparing $M_u$ and $M_v$, and $\mu'_{u,v}$ denotes the mean similarity of off-diagonal cells. A naive approach to generate a $p$-value for $H_0^{u,v}$, which assumes that the affinity matrix's entries are independent, is to perform a one-sided $t$-test comparing on- and off-diagonal cells in the affinity matrix for $M_u$ and $M_v$. Performing such a $t$-test for each model pair, we find that the resulting $p$-values are astronomically small, with a maximum of $8 \times 10^{-7}$. Moreover, the $p$-value for the union null hypothesis

$$H_0 = \bigcup_{u,v} H_0^{u,v} : \exists_{u,v} \ \mu_{u,v} \leq \mu'_{u,v}$$

is given by the maximum of all of the constituent $p$-values (Berger, 1982; Kim & Allison, 2006). Therefore, according to the naive $t$-tests, the $p$-value for $H_0$ is $8 \times 10^{-7}$, indicating that *every pair* of models exhibit statistically higher similarity on the diagonal than off of it.

**Block bootstrap** The $t$-test approach assumes independence of the cells in each affinity matrix. However, this assumption is clearly violated, as any pair of cells in a row $i$—e.g., $A_{i,j}^{u,v}$ and $A_{i,j'}^{u,v}$—for the affinity matrix comparing models $M_u$ and $M_v$ are both functions of the same embedding function $f_{M_u}^{(i)}$. The same is true for pairs of cells in any column $j$, both of which are functions of $f_{M_v}^{(j)}$. In addition, the embedding functions at nearby layers within a model—e.g., $f_{M_u}^{(i)}$ and $f_{M_u}^{(i+1)}$—may be dependent due to residual connections and other aspects of the transformer architecture.

To account for these local dependencies, we use the moving block bootstrap (Lahiri, 1999; Davison & Hinkley, 1999) with a block size of $5 \times 5$ to sample from a null distribution which preserves local dependencies but scrambles diagonal structure (see Appendix B for more details). By comparing samples from this null distribution to the actual affinity matrix, we can estimate the probability that the difference of on- and off-diagonal means would be as great under the null model as those of the observed affinity matrix. Repeating this process for every affinity matrix, we find a maximum $p$-value of $4.84 \times 10^{-3}$, again indicating that the on-diagonal mean is greater than the off-diagonal mean for *every pair* of models.

## 5 Conclusion

We compare the activations of a wide range of modern LLMs and find that they generate different activation geometries at different depths (Claim 1), but that layers at corresponding depths from different models tend to generate similar activation geometries (Claim 2). We find that early layers from different models are just as similar to one another as middle and late layers, and validate our findings by developing a statistical test of diagonal structure.

## Acknowledgments

We would like to thank Claire Donnat and Nikos Ignatiadis for their advice on statistical tests of diagonal structure.

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

## A   Diagonal structure is sensitive to the input distribution

The experiments above all focused on the similarity of nearest neighbor relationships on a dataset of inputs from OpenWebText. In Appendix G, we generate affinity matrices for a range of input datasets, including IMDB movie reviews (Maas et al., 2011), text from books in English and German (Tiedemann, 2012), and various LLM evaluations (Zhou et al., 2023; Hendrycks et al., 2021). The diagonal structure observed in Section 4 is generally present for other distributions of web text (e.g., IMDB movie reviews), but different patterns emerge when looking at other input distributions, which we investigate here.

### A.1   Nearest neighbor relationships are (weakly) preserved between languages

Many of the models we use in our experiments have been trained on multilingual data, and can write and understand multiple languages. In the experiments above, we feed the same dataset $\mathcal{D}$ to two models $M_1$ and $M_2$ and collect the resulting activations. Here, using a parallel corpus of English and German book translations (Tiedemann, 2012), we instead feed a corpus of English texts to $M_1$ and a parallel corpus of translated German texts to $M_2$. The resulting nearest neighbor sets exhibit lower similarity than when using only a single language, but we still observe weak diagonal structure, with middle layers being more similar to one another than to early or late layers (Figure 7).

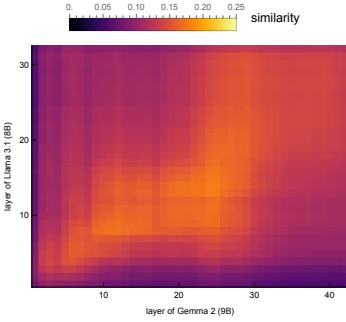

Figure 7: Weak diagonal structure is present when comparing embeddings of different models on parallel texts in different languages. Note that the color scale is zoomed to show lower similarity cells better.

### A.2   Instruction tuning changes activation structure in late layers

Instruction tuning is a process in which a base model (which has generally been pretrained on natural language) is fine-tuned to be better at following instructions (Zhang et al., 2024). We investigate how instruction tuning affects the activations generated by language models.

We compare the base version of Gemma-2-9B to its instruction-tuned counterpart. When given inputs from OpenWebText, we observe strong diagonal structure across all depths (Figure 8a). However, when given prompts from a dataset of inputs designed to test a model's ability to follow instructions (Zhou et al., 2023) there is much lower similarity at later layers (Figure 8b). This finding is in line with previous work that observed later layers being more strongly affected by fine-tuning (Wu et al., 2020).

We hypothesize that later layers could be dedicated to functions that are strongly targeted by instruction tuning, while early layers could be dedicated to functions like input parsing and basic grammar that are needed with or without instruction tuning. Alternatively, the instruction tuning process used could have been designed to only train later layers.

Regardless of the mechanism, this shows that 1) the distribution from which input texts are drawn can affect diagonal structure, and 2) that we can study which parts of a model have been affected by fine-tuning with affinity matrices.

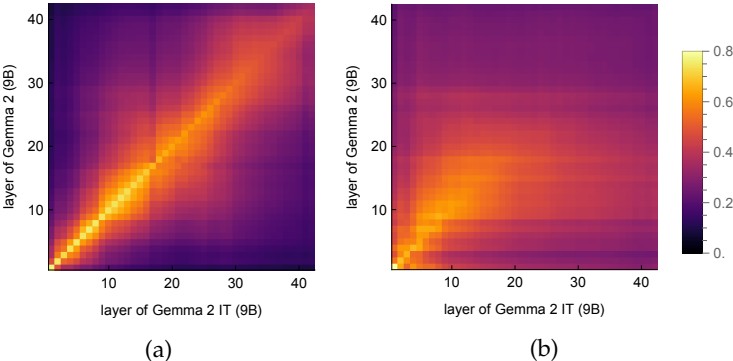

(a)                                          (b)

Figure 8: Instruction tuning changes the activations generated by models in later layers, but only when given inputs that are targeted by instruction tuning. 8a The affinity matrix of the base version of `Gemma-2-9B` and its instruction tuned counterparts, given a dataset of inputs from OpenWebText. 8b The same as 8a, but using a dataset of inputs from IFEval.

### A.3    Random alphanumeric strings

We additionally test sensitivity to the input distribution by repeating the experiments above using a dataset of random alphanumeric strings. Specifically, we generate 2048 strings of length 100, each consisting of letters, numbers, and punctuation sampled uniformly at random.

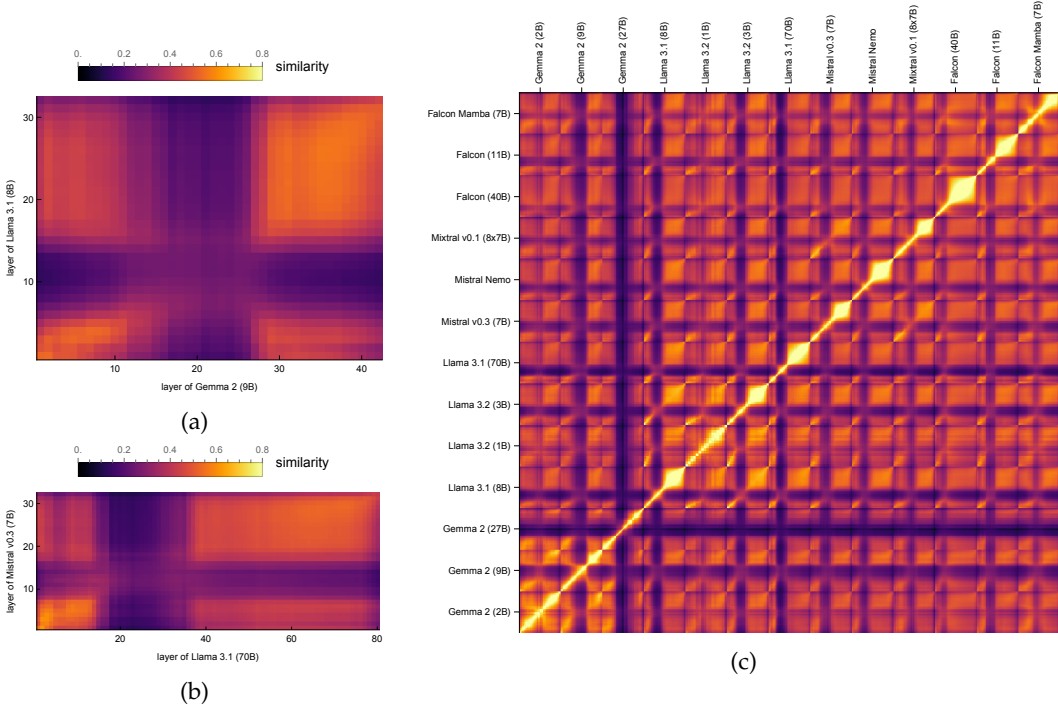

Figure 9: Affinity matrices generated with a dataset of random strings reveal similarity in nearest neighbor relationships, but not diagonal structure.

Figure 9 reveals that there is similarity between the nearest neighbor relationships of random strings, but *without diagonal structure*. Just like with web text in Section 4, layers from different models often agree on the nearest neighbor relationships among the random strings. However, unlike with web text, the most similar layers do not necessarily come

from corresponding depths. Instead, all early and late layers show similarity with one another, while middle layers are dissimilar between models.

Further investigation of the nearest neighbor sets among random strings reveals that early and late layers tend to assign nearest neighbors based on agreement in the last few characters. For example, if a string ends with "Uc", then early and late layers will generally consider any other strings ending with "Uc" to be a nearest neighbor. This leads to high agreement, as all of these layers are focused on the superficial feature of the inputs. That middle layers do not follow this pattern could suggest that they are not as susceptible to focusing on superficial features of the input, and so generate less structured activations when given random strings.

This experiment shows that, at least in extreme cases, the diagonal structure observed above is contingent on the input distribution. It also shows that the mere presence of similarity between two models does not tell the full story, and that we learn more from looking at the *pattern* of similarity between different pairs of layers.

## B  Block bootstrap test of diagonal structure

Given an affinity matrix, we want to test whether the on-diagonal cells show higher similarity scores than the off-diagonal cells. Naively, we could perform a one-sided $t$-test comparing the scores in on- and off-diagonal cells, but this would assume that each cell is independent, which need not be true because of residual connections in transformer models (Vaswani et al., 2017).

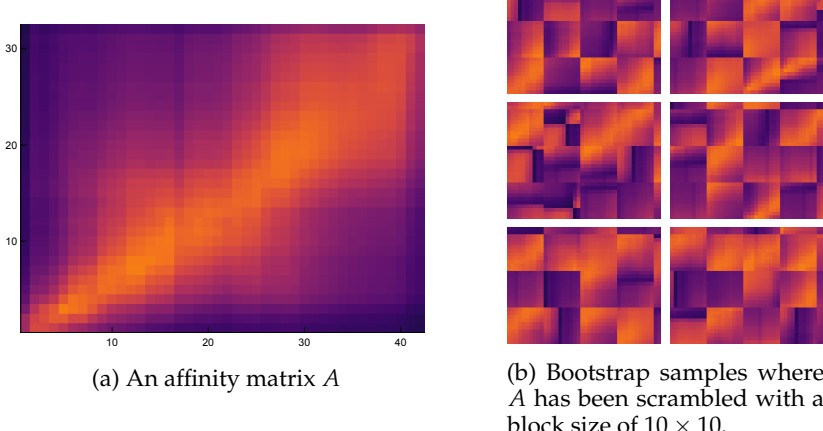

(a) An affinity matrix $A$

(b) Bootstrap samples where $A$ has been scrambled with a block size of $10 \times 10$.

Figure 10

We use the block bootstrap to generate samples from a null distribution that preserves local dependence structure, but scrambles large-scale structure (Lahiri, 1999; Davison & Hinkley, 1999). The block bootstrap works by breaking an affinity matrix into blocks of a given size and then resampling those blocks (with replacement) to create a new matrix that inherits the local dependence of the original, but does not inherit its global structure (Figure 10). We use overlapping blocks with a cyclic boundary condition so that every cell has an equal probability of appearing in the resampled matrix. By comparing the difference of on- and off-diagonal means in the observed affinity matrices with their corresponding resampled matrices, we test whether the observed diagonal structure is stronger than would be expected as a consequence of the local dependence structure and chance.

For each affinity matrix, we generate $10^5$ scrambled bootstrap samples and compute the difference of the on- and off-diagonal means. The fraction of bootstrap samples that yields a difference at least as large as that observed from the original affinity matrix gives us a $p$-value (Figure 11).

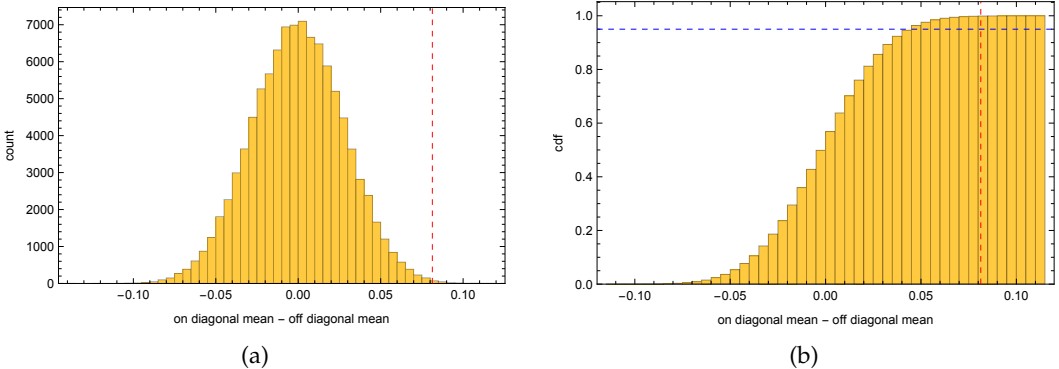

Figure 11: Distribution of differences between on- and off-diagonal means for bootstrap samples for a particular affinity matrix. The red line shows the difference between on- and off-diagonal means observed with the unscrambled affinity matrix. In Figure 11b, the blue line shows 0.95.

We used this process to compute a *p*-value for every affinity matrix (generated on Open-WebText data), using block sizes of $5 \times 5$ and $10 \times 10$. For most model pairs, there was not a single bootstrap sample with an on- and off-diagonal difference as large as the observed affinity matrix (Figure 12). No model pair yielded a *p*-value greater than 0.05 with either block size. The maximum *p*-value with block size $5 \times 5$ is 0.00484, and the maximum *p*-value with block size $10 \times 10$ is 0.0154.

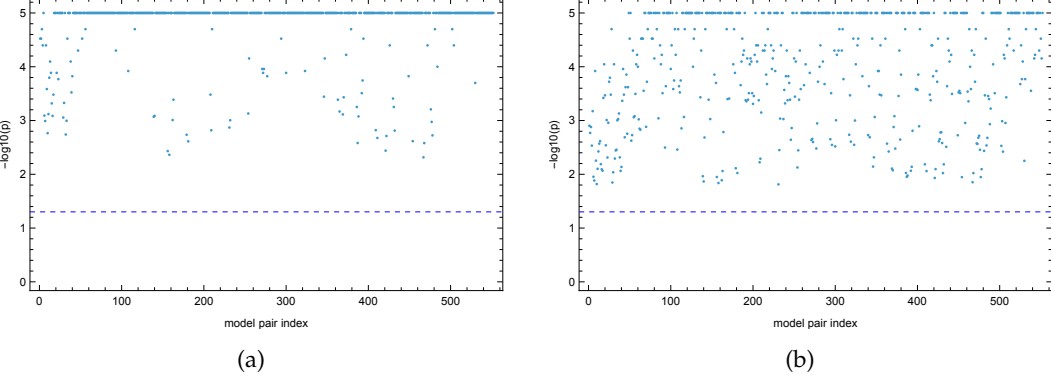

Figure 12: Every test yields a *p*-value below 0.05. 12a Manhattan plot generated with a block size of $5 \times 5$. Tests where no bootstrap sample had a difference as great as the observed difference are clipped to one over the number of bootstrap samples for visualization purposes. 12b Manhattan plot generated with a block size of $10 \times 10$.

We can combine these tests into a single intersection-union test of whether the on-diagonal similarity is higher than the off-diagonal similarity for every pair of models simultaneously. Because we can reject the null-hypothesis for each of the individual tests (as the maximum *p*-value is below 0.05), we can reject it for the intersection-union test, giving us the strong result that *every pair* of models shows significant diagonal structure (Berger, 1982; Kim & Allison, 2006).

## C Null model for mutual $k$-nearest neighbors

How can we interpret the absolute values of mutual $k$-NN similarity scores, and how much similarity would be expected by random chance?

**Deriving the null model** Suppose that the nearest neighbor function $\mathcal{U}_k^{(\mathcal{D})}$ returned every size $k$ subset of $\mathcal{D}$ with equal probability, i.e.

$$\mathcal{U}_k^{(\mathcal{D})}(x, f) \sim \text{Uniform}(\{X \mid X \subset \mathcal{D}, \|X\| = k\}) \tag{3}$$

for any $x$ and $f$. We then have that

$$\left\|\mathcal{U}_k^{(\mathcal{D})}(x, f) \cap \mathcal{U}_k^{(\mathcal{D})}(x, g)\right\| \sim \text{Hypergeometric}(k, k, \|\mathcal{D}\| - 1) \tag{4}$$

where $\text{Hypergeometric}(k, k, \|\mathcal{D}\| - 1)$ is the distribution of successes in $k$ draws from a population of size $\|\mathcal{D}\| - 1$ that contains $k$ successes. By the central limit theorem, we then have that

$$\mathcal{M}_k^{(\mathcal{D})}(f, g) = \frac{1}{k\|\mathcal{D}\|} \sum_{x \in \mathcal{D}} \left\|\mathcal{U}_k^{(\mathcal{D})}(x, f) \cap \mathcal{U}_k^{(\mathcal{D})}(x, g)\right\| \tag{5}$$

$$\sim \text{Normal}\left(\frac{k}{\|D\| - 1}, \sqrt{\frac{(k - \|\mathcal{D}\| + 1)^2}{\|\mathcal{D}\|(\|\mathcal{D}\| - 2)(\|\mathcal{D}\| - 1)^2}}\right) \tag{6}$$

which gives the distribution of $\mathcal{M}_k^{(\mathcal{D})}(f, g)$ under the null assumption made above.

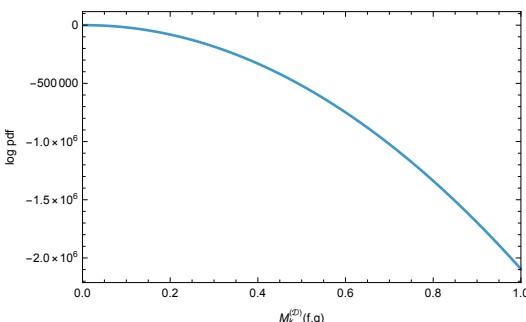

Figure 13: Null distribution of $\mathcal{M}_k^{(\mathcal{D})}(f, g)$ with $k = 10$ and $\|\mathcal{D}\| = 2048$. The log pdf is astronomically small for values larger than approximately 0.02.

**Applying the null model** The null distribution derived above is extremely concentrated, with the null probability of $\mathcal{M}_k^{(\mathcal{D})}(f, g) \geq 0.4$ being on the order of $10^{-143,449}$ (when $k = 10$ and $\|\mathcal{D}\| = 2048$). Thus, the mutual $k$-NN similarity scores appearing in the affinity matrices above are far higher than would be expected under this (very strong) null model.

At the same time, we emphasize the pattern of similarity more so than its absolute value. Even if the level of similarity were much lower, it would still be remarkable that there is *more* similarity among layers of similar depths when compared with pairs of layers with different depths.

# D  Other values of $k$

In the experiments above, we measure similarity with mutual $k$-NN with $k = 10$. Here, we generate versions of Figure 4a using various other values of $k$ (Figure 14). The results show similar diagonal structure, even with $k = 1$ (which corresponds to the probability that the layers being compared agree on the single nearest neighbor).

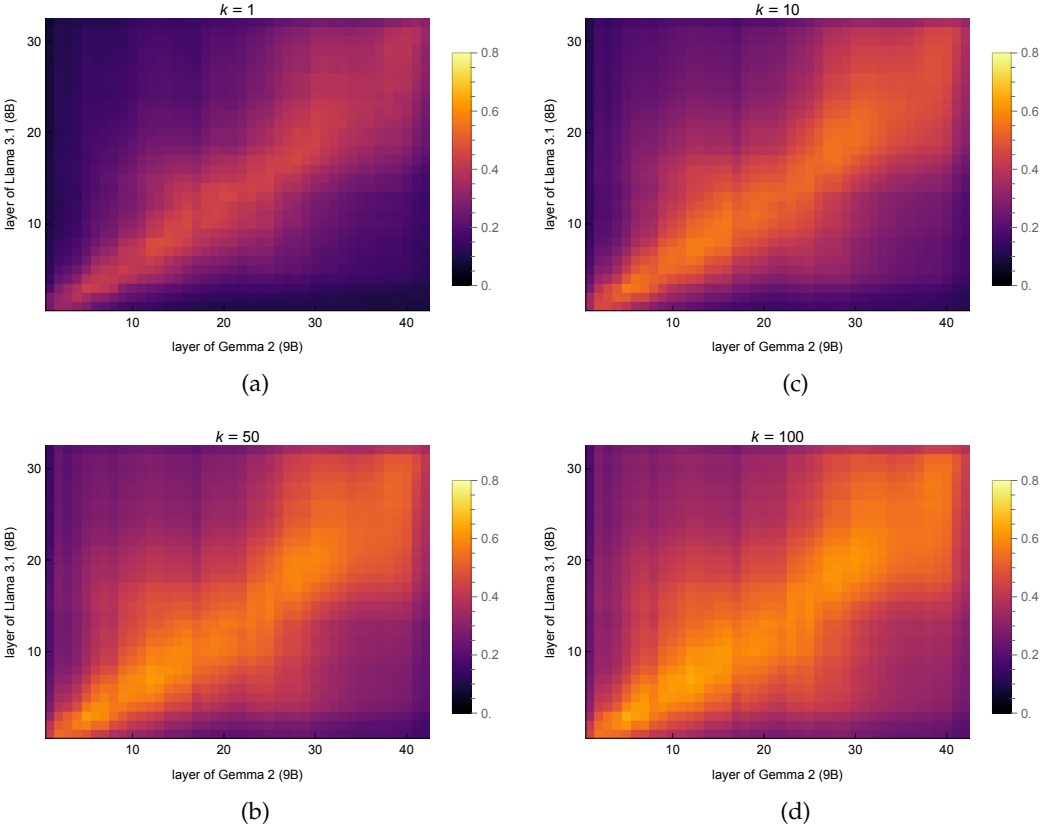

Figure 14: Affinity matrices comparing activations of OpenWebText from layers of `Llama-3.1-8B` and `Gemma-2-9B` with different values of $k$. Diagonal structure is present throughout.

# E   Decay curves

The following plots visualize how the relative depth of layers relates to their similarity. In particular, they show how similarity decays as distance increases.

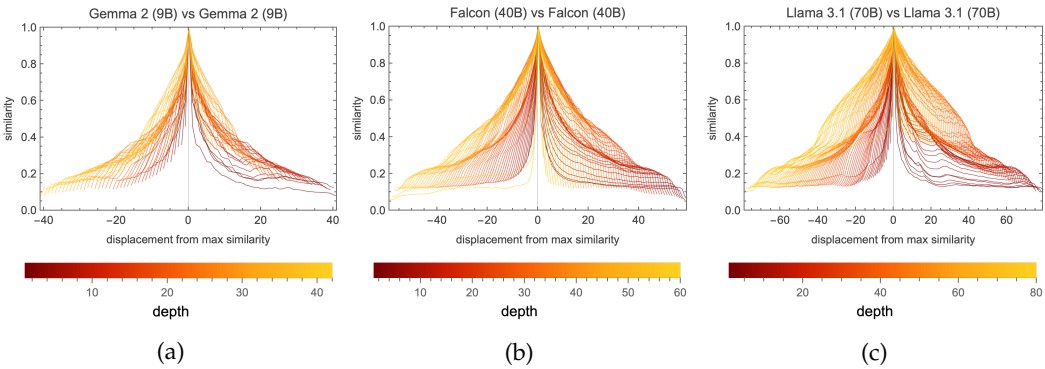

Figure 15: Nearby layers within a model tend to be more similar than distant ones. For a given model, each plot shows the similarity between layers as a function of their distance. Each line corresponds to a layer with the color indicating its depth. The *x*-axis shows the relative depth of another layer in the same model, and the *y*-axis shows their similarity. Each curve can be thought of as showing a slice of the affinity matrices above. All plots use mutual *k*-nearest neighbors with OpenWebText as the dataset.

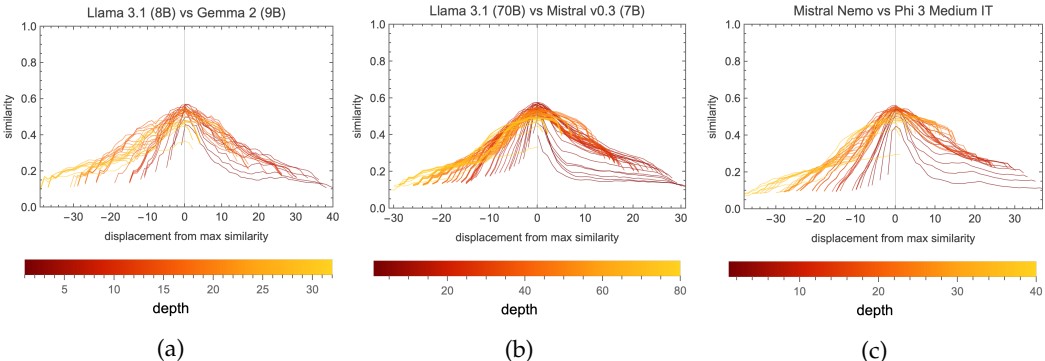

Figure 16: Corresponding layers from different models tend to be more similar than non-corresponding ones. For a given pair of models, each plot shows the similarity between layers in one model to layers in another model as those layers get further away from the path of maximum similarity. Each line corresponds to a layer in one model with the color indicating its depth. The *x*-axis shows the depth of the layer in the other model, shifted so that the highest-similarity layer is at 0. The *y*-axis shows the similarity of the two layers. Each curve can be thought of as showing a slice of the affinity matrices above. All plots use mutual *k*-nearest neighbors with OpenWebText as the dataset.

# F   Model list

We use the following models in our experiments:

| Name | Layers | IT | HuggingFace ID |
|---|---|---|---|
| Gemma (2B) | 18 | No | google/gemma-2b |
| Gemma (7B) | 28 | No | google/gemma-7b |
| Gemma 2 (2B) | 26 | No | google/gemma-2-2b |
| Gemma 2 (9B) | 42 | No | google/gemma-2-9b |
| Gemma 2 IT (9B) | 42 | Yes | google/gemma-2-9b-it |
| Gemma 2 (27B) | 46 | No | google/gemma-2-27b |
| Llama 3.1 (8B) | 32 | No | meta-llama/Meta-Llama-3.1-8B |
| Llama 3.1 IT (8B) | 32 | Yes | meta-llama/Meta-Llama-3.1-8B-Instruct |
| Llama 3.2 (1B) | 16 | No | meta-llama/Llama-3.2-1B |
| Llama 3.2 (3B) | 28 | No | meta-llama/Llama-3.2-3B |
| Llama 3.2 IT (3B) | 28 | Yes | meta-llama/Llama-3.2-3B-Instruct |
| Llama 3.2 Vision (11B) | 40 | No | meta-llama/Llama-3.2-11B-Vision |
| Llama 3.1 (70B) | 80 | No | meta-llama/Llama-3.1-70B |
| Llama 3.1 IT (70B) | 80 | Yes | meta-llama/Llama-3.1-70B-Instruct |
| Llama 3.3 IT (70B) | 80 | Yes | meta-llama/Llama-3.3-70B-Instruct |
| Mistral v0.3 (7B) | 32 | No | mistralai/Mistral-7B-v0.3 |
| Mistral Nemo | 40 | No | mistralai/Mistral-Nemo-Base-2407 |
| Mixtral v0.1 (8x7B) | 32 | No | mistralai/Mixtral-8x7B-v0.1 |
| Phi 3 Mini IT | 32 | Yes | microsoft/Phi-3-mini-4k-instruct |
| Phi 3 Medium IT | 40 | Yes | microsoft/Phi-3-medium-4k-instruct |
| Phi 3.5 Mini IT | 32 | Yes | microsoft/Phi-3.5-mini-instruct |
| Falcon (40B) | 60 | No | tiiuae/falcon-40b |
| Falcon (11B) | 60 | No | tiiuae/falcon-11B |
| Falcon Mamba (7B) | 64 | No | tiiuae/falcon-mamba-7b |

Table 1: Models used in experiments, their layer counts, and whether they were instruction tuned (IT).

# G   Extended affinity matrix arrays

We computed affinity matrices comparing all pairs of all layers in 24 open-weight LLMs. For the sake of brevity, we only included a subset in Figure 4c. This appendix contains affinity matrices for all models on all datasets that we tested.

## G.1   Rectangular affinity matrices

These affinity matrices have not been resized to fit into squares, and preserve the original dimensions of the affinity matrices, with deeper models occupying more space.

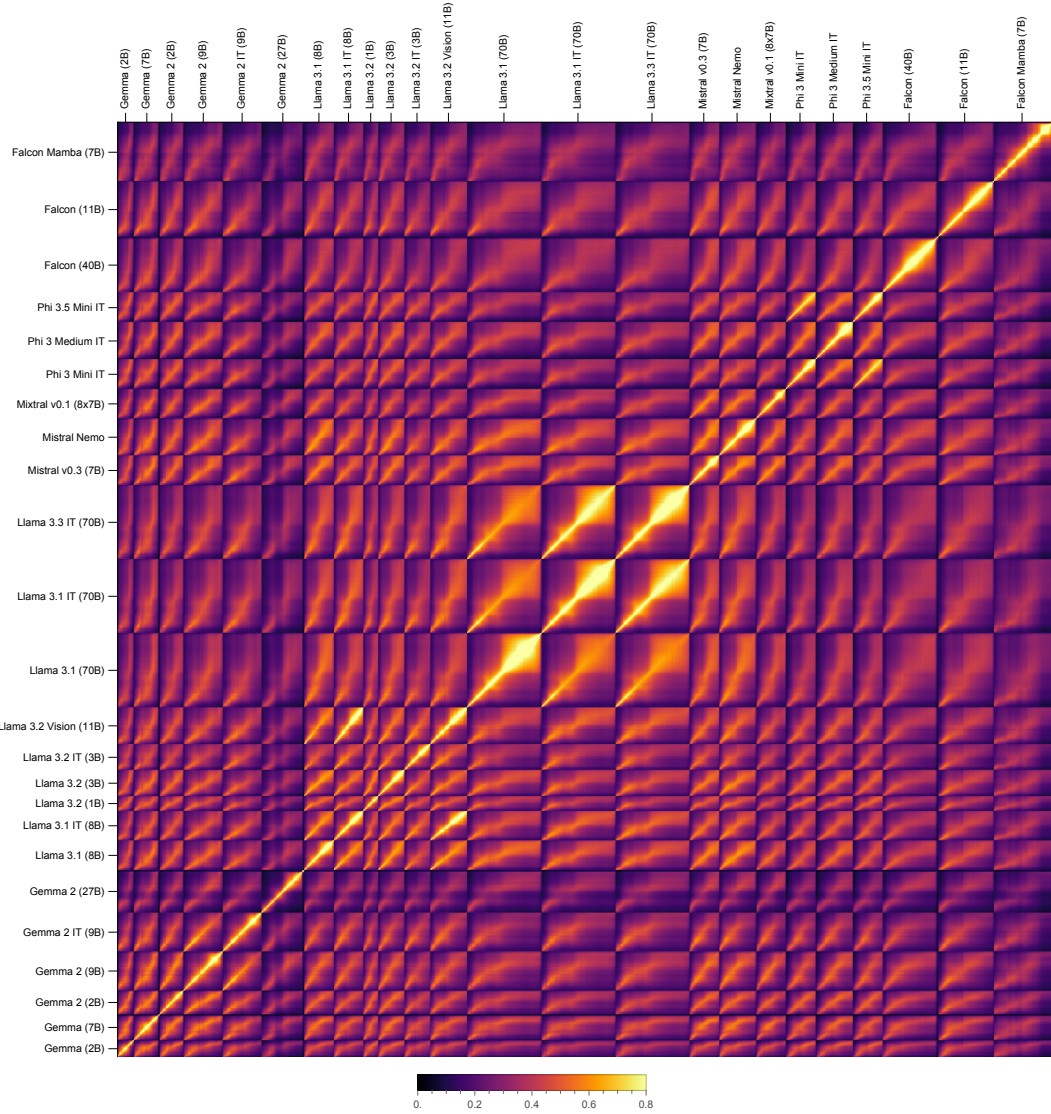

Figure 17: Affinity matrices for embeddings of OpenWebText (Gokaslan et al., 2019).

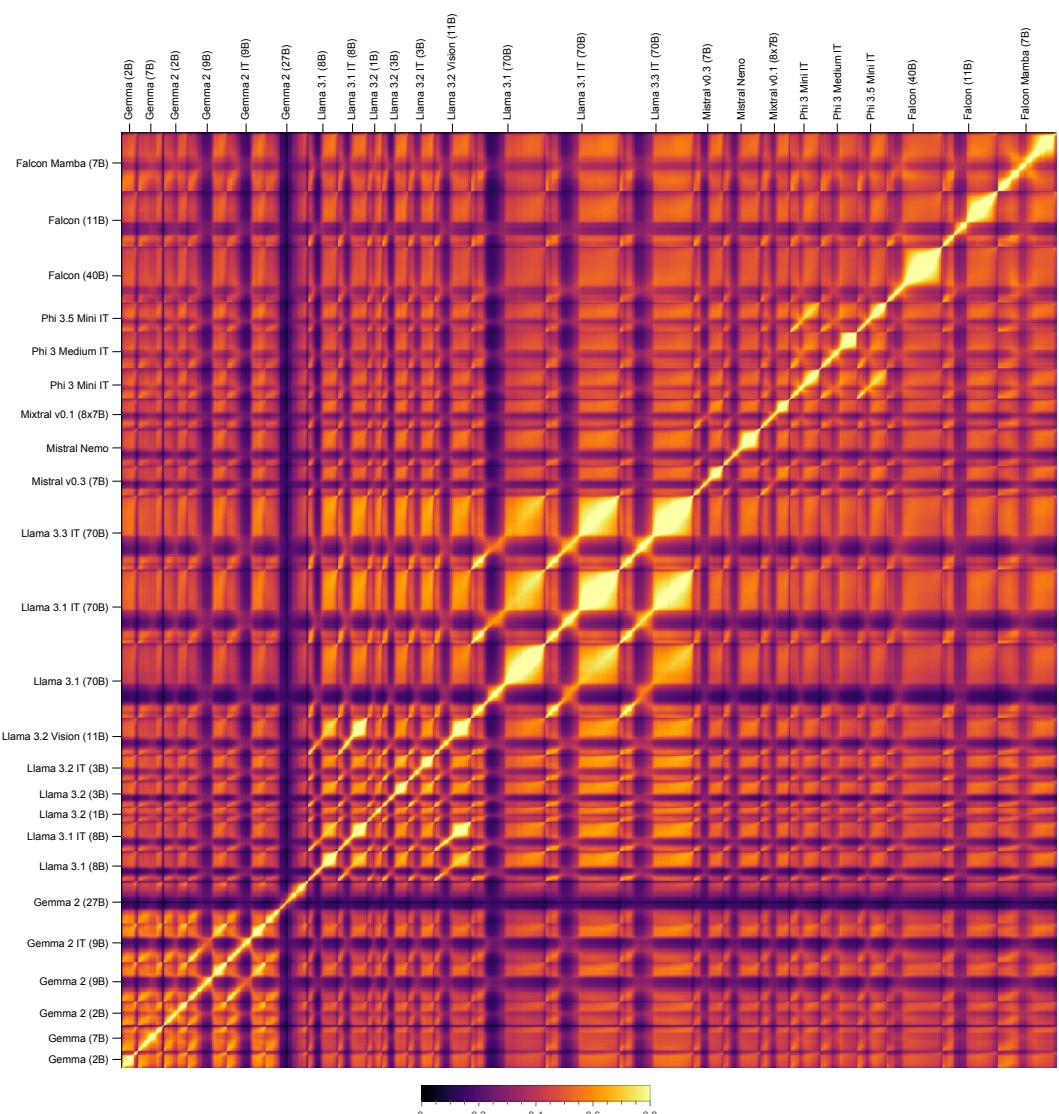

Figure 18: Affinity matrices for embeddings of random alphanumeric strings.

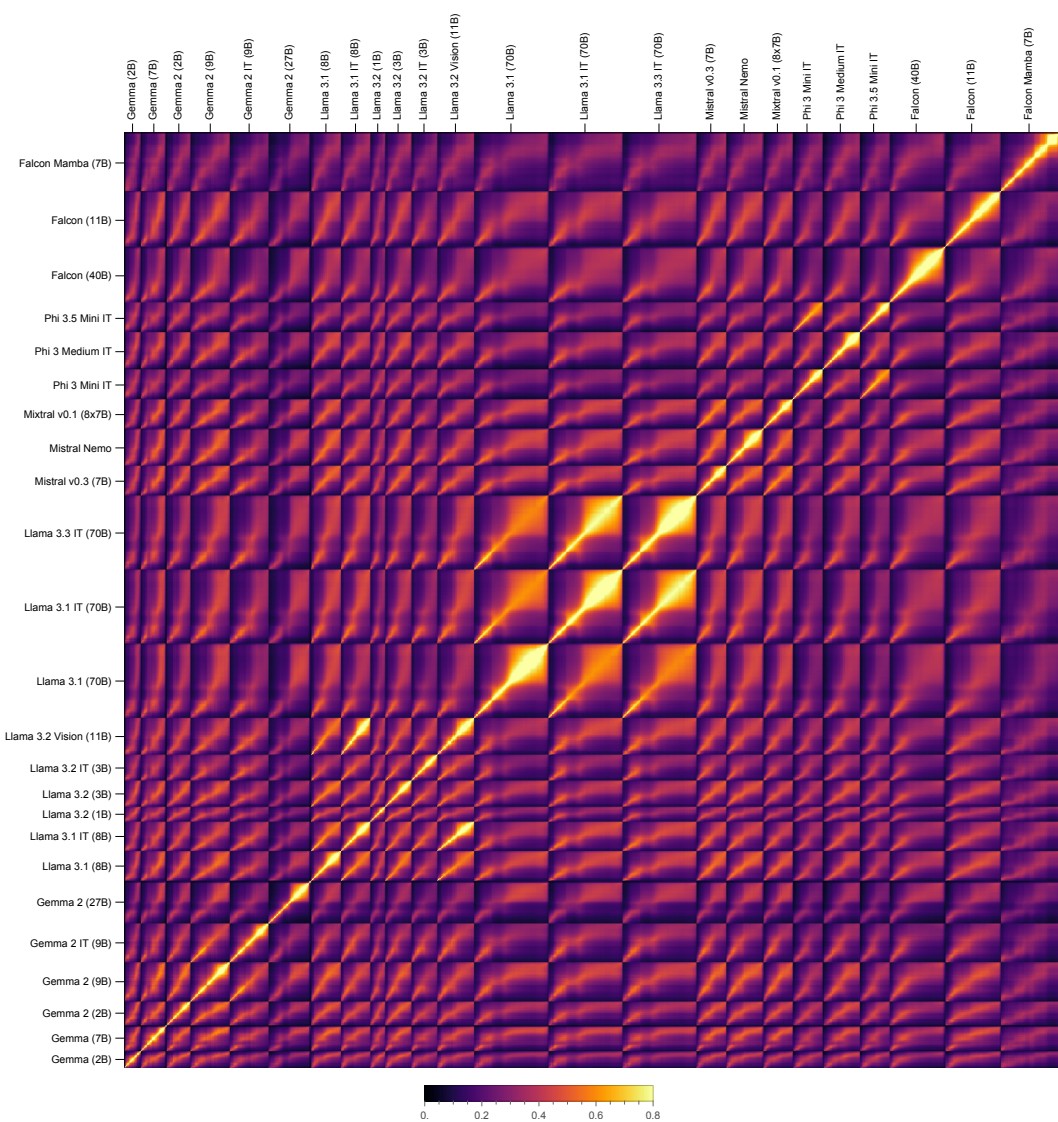

Figure 19: Affinity matrices for embeddings of IMDB movie reviews (Maas et al., 2011).

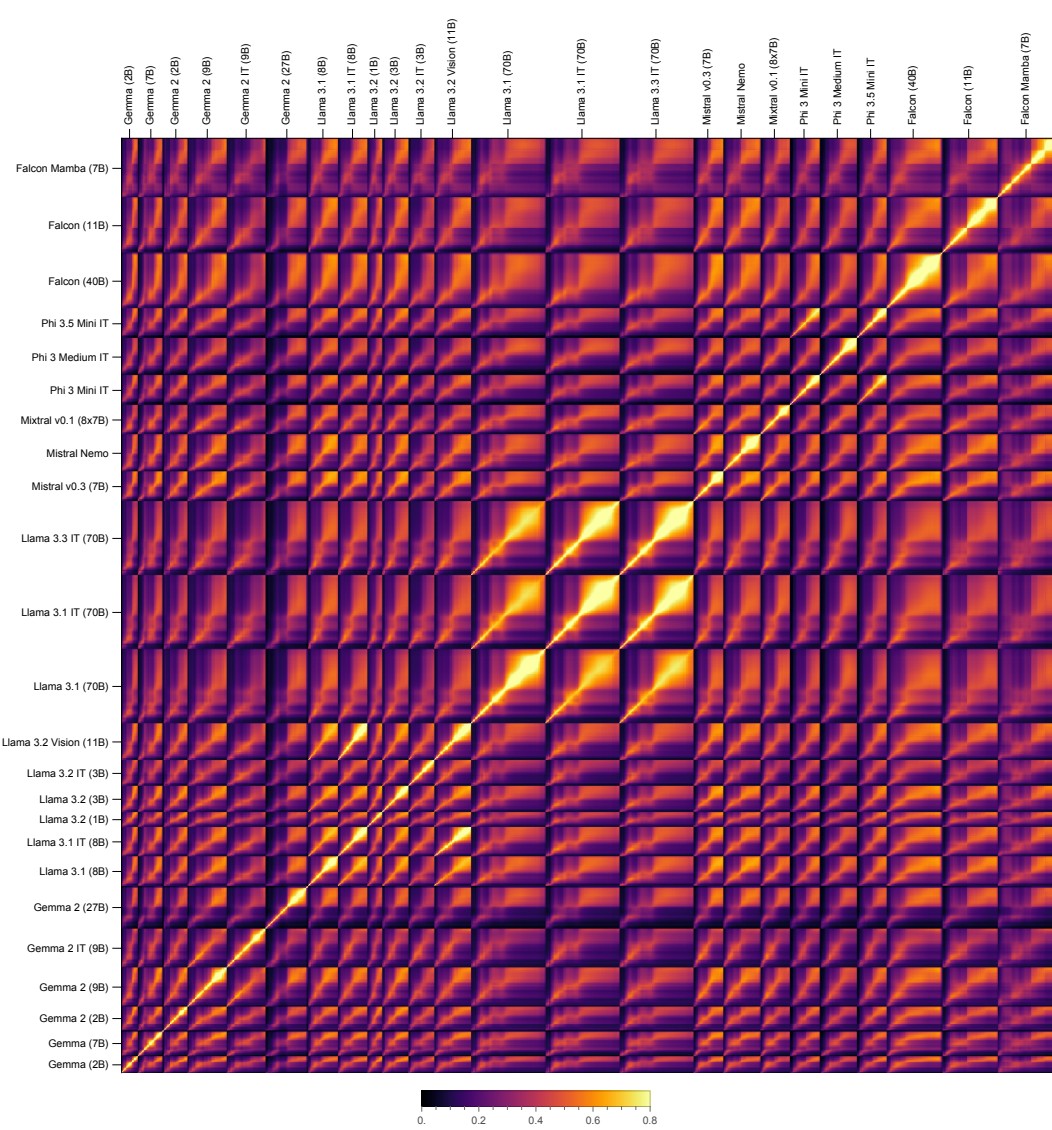

Figure 20: Affinity matrices for embeddings of lead paragraphs from featured articles on Wikipedia.

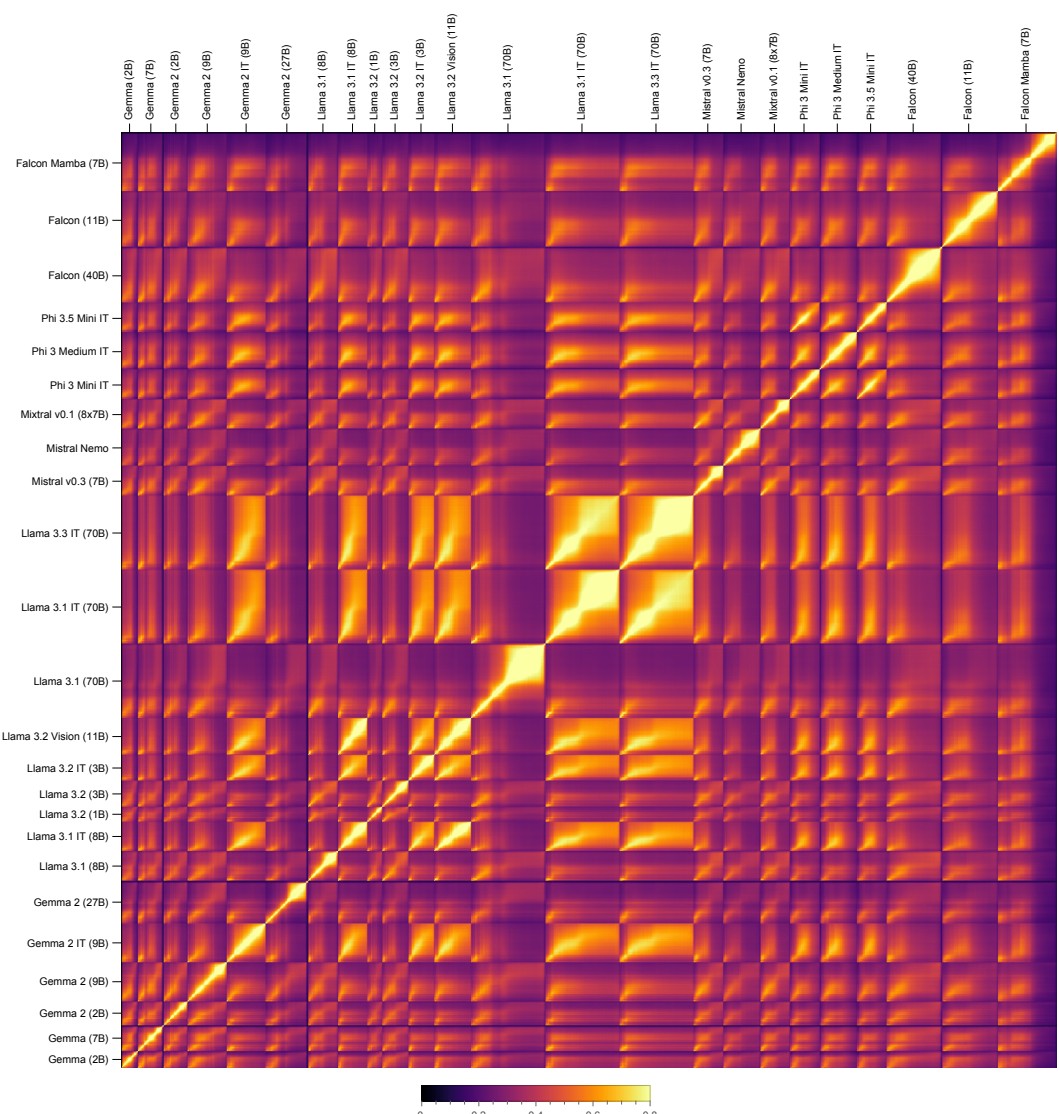

Figure 21: Affinity matrices for embeddings of IFEval (Zhou et al., 2023).

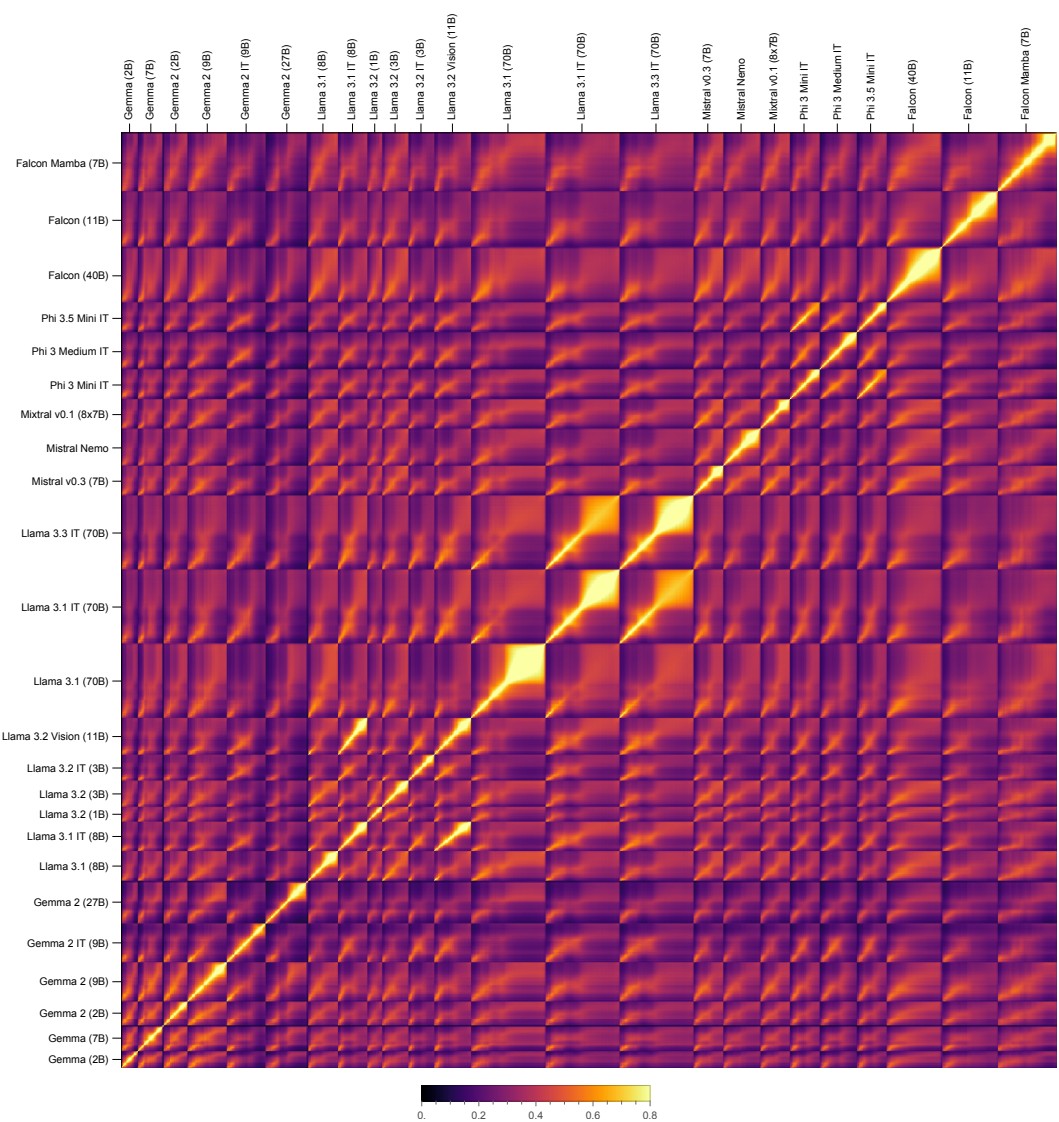

Figure 22: Affinity matrices for embeddings of MMLU (Hendrycks et al., 2021).

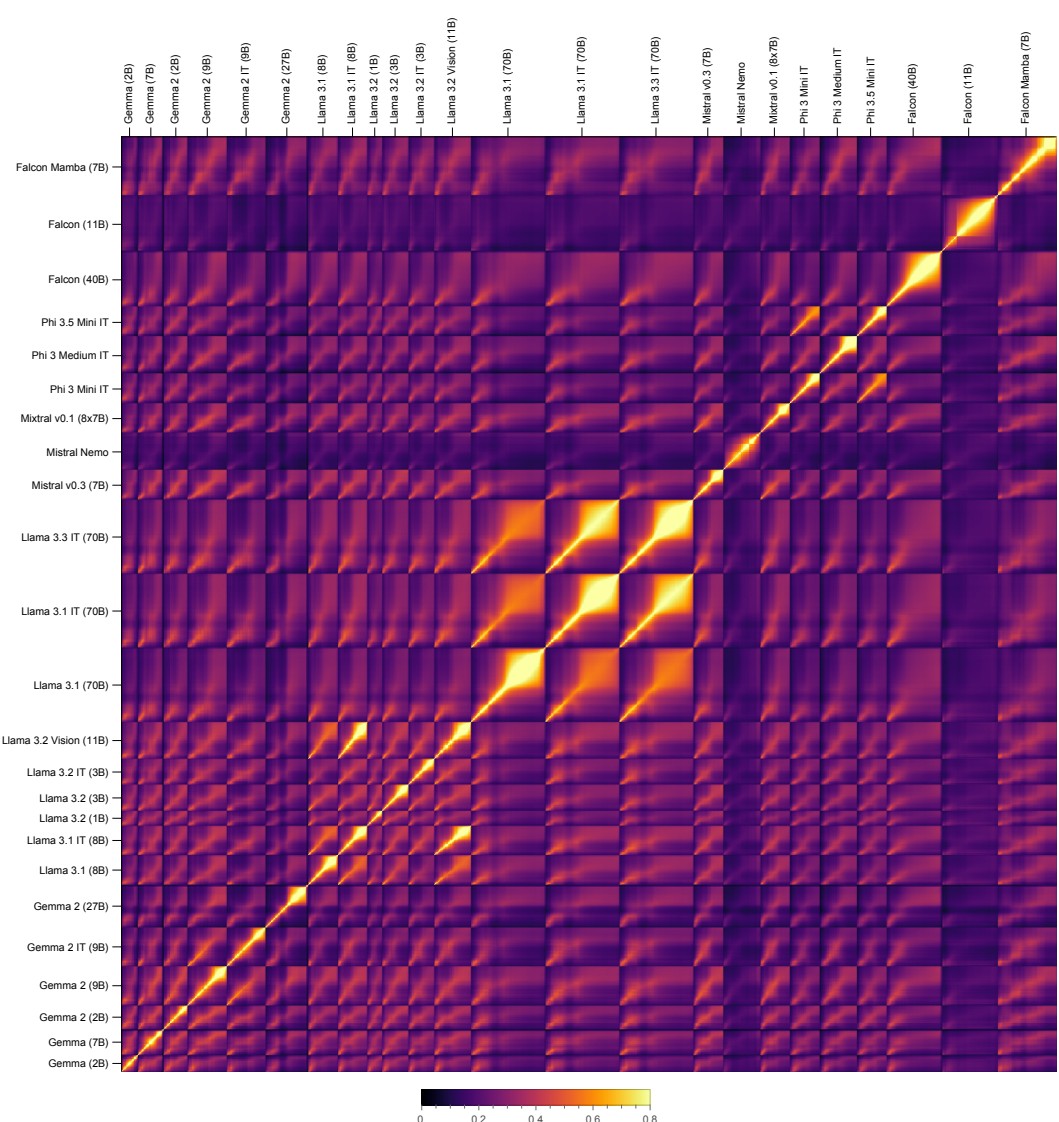

Figure 23: Affinity matrices for embeddings of OPUS Books (English) (Tiedemann, 2012).

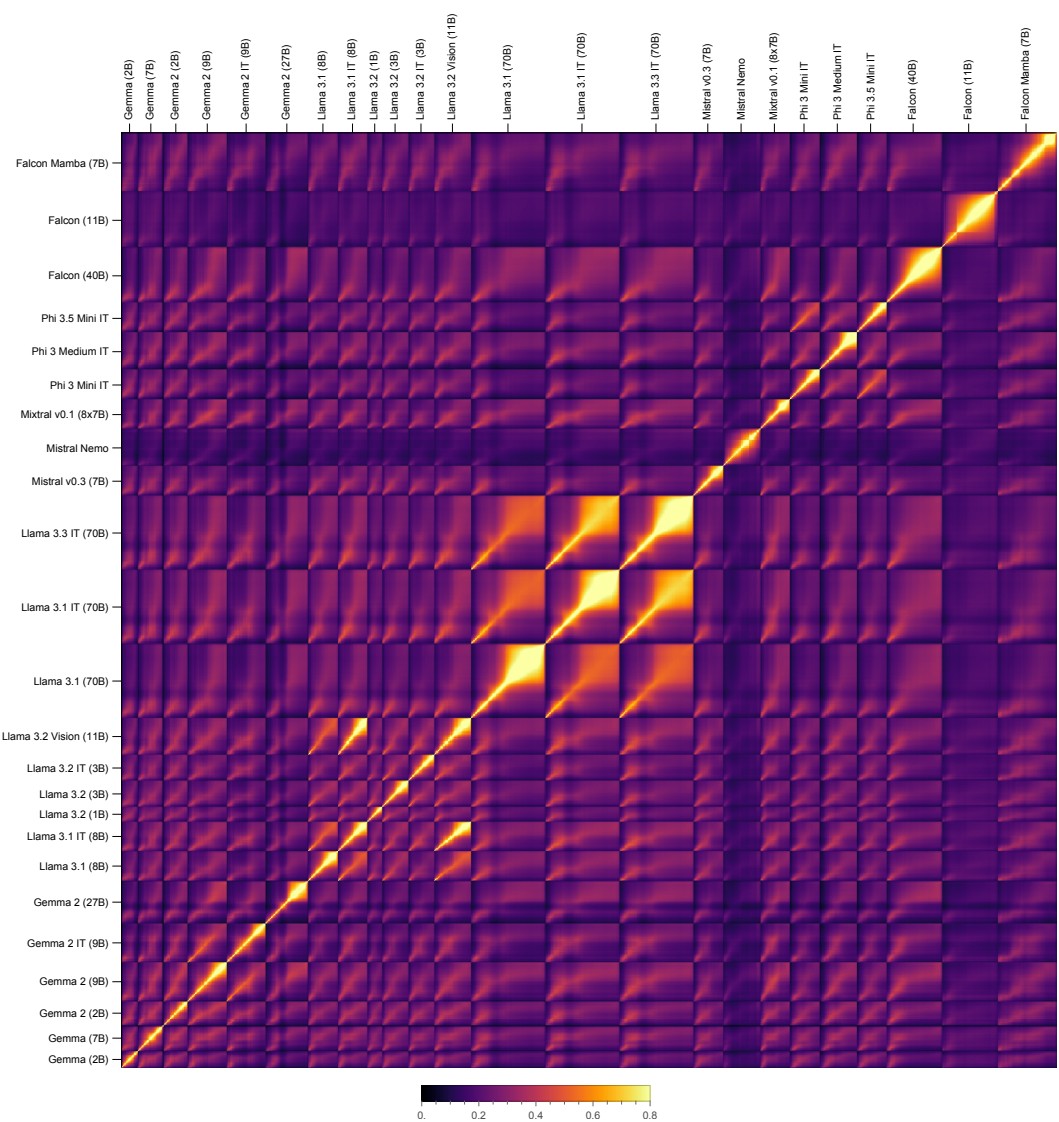

Figure 24: Affinity matrices for embeddings of OPUS Books (German) (Tiedemann, 2012).

## G.2 Square affinity matrices

These affinity matrices have been stretched and squished to fit into squares to make them more comparable, as in Figure 4c.

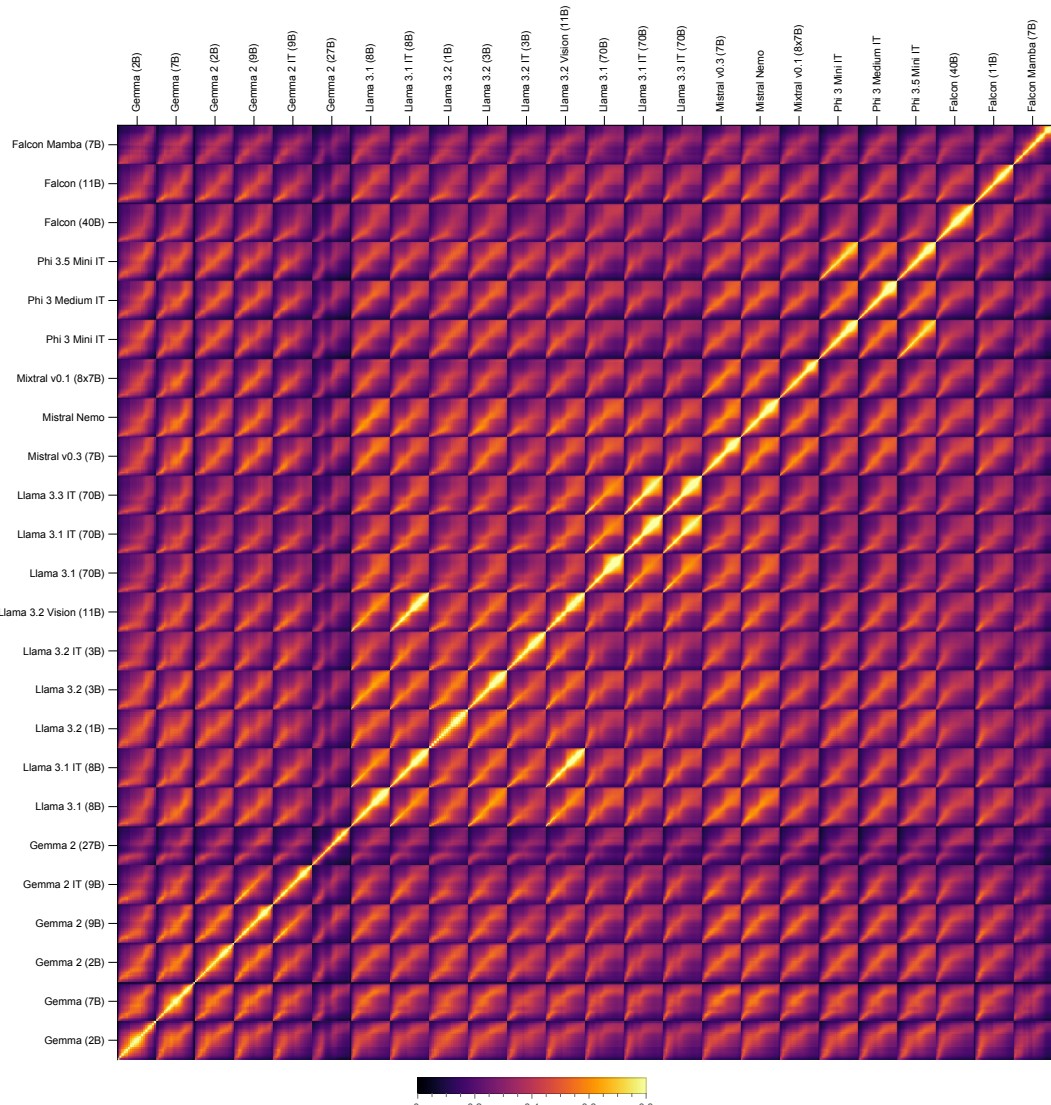

Figure 25: Affinity matrices for embeddings of OpenWebText (Gokaslan et al., 2019).

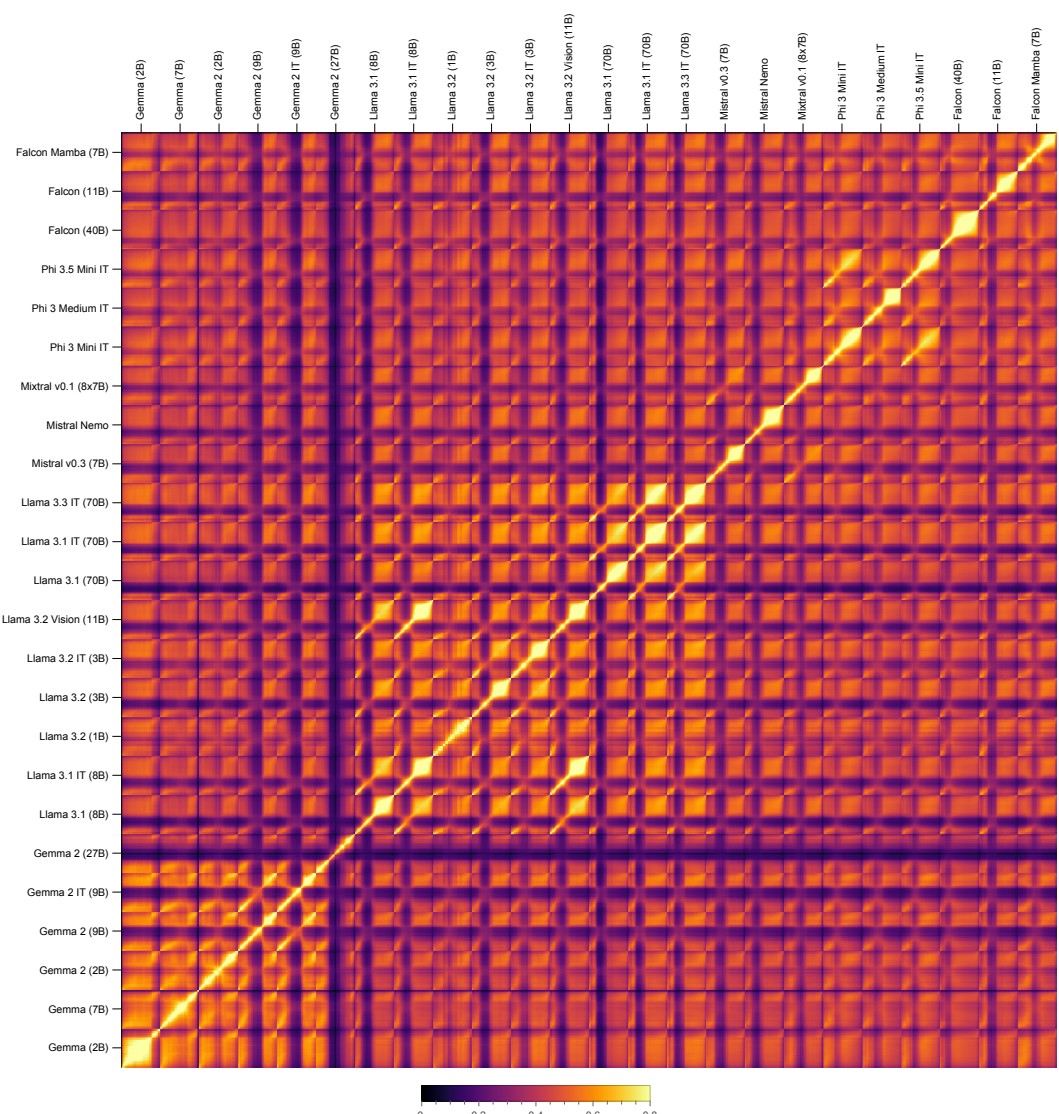

Figure 26: Affinity matrices for embeddings of random alphanumeric strings.

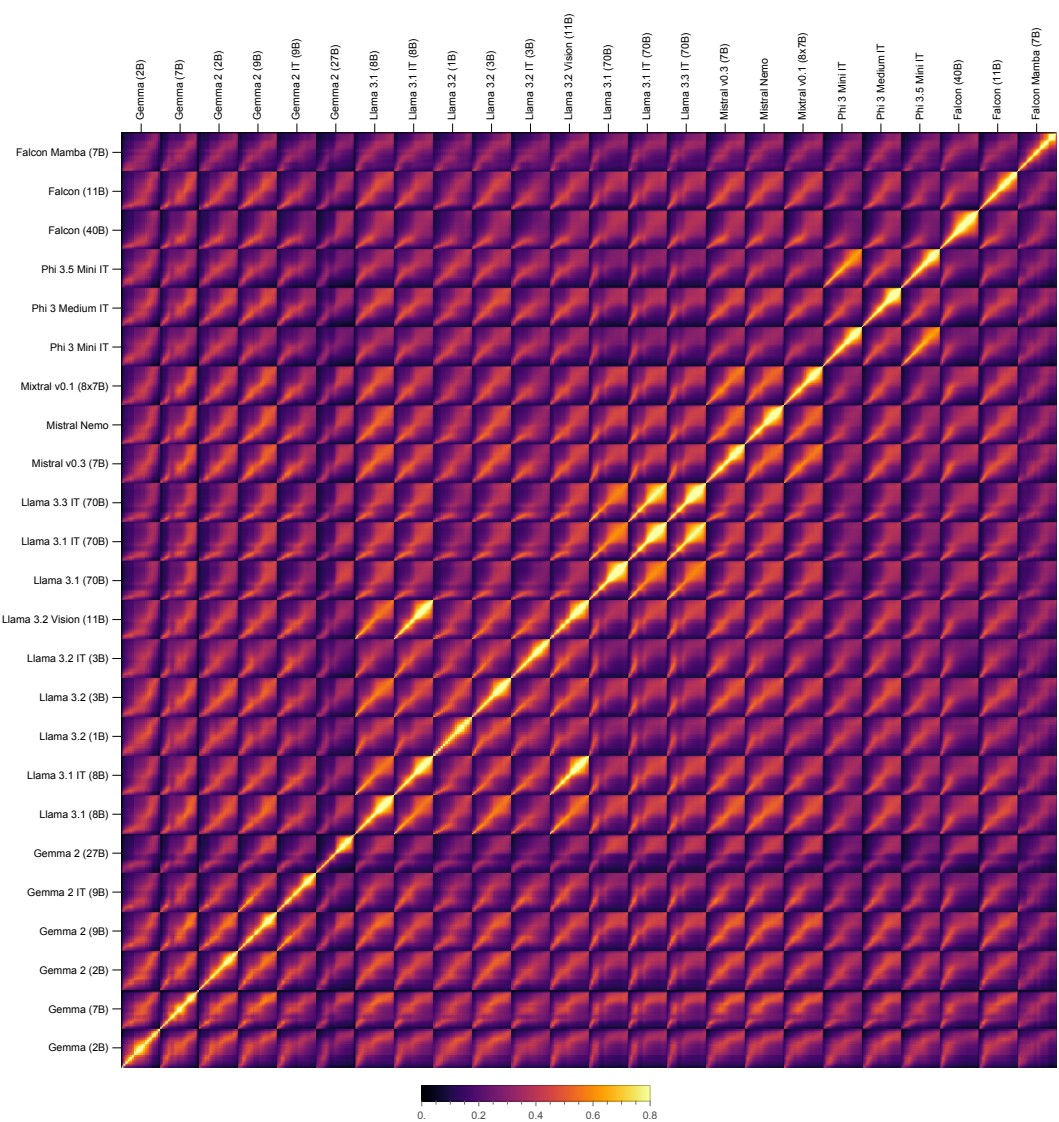

Figure 27: Affinity matrices for embeddings of IMDB movie reviews (Maas et al., 2011).

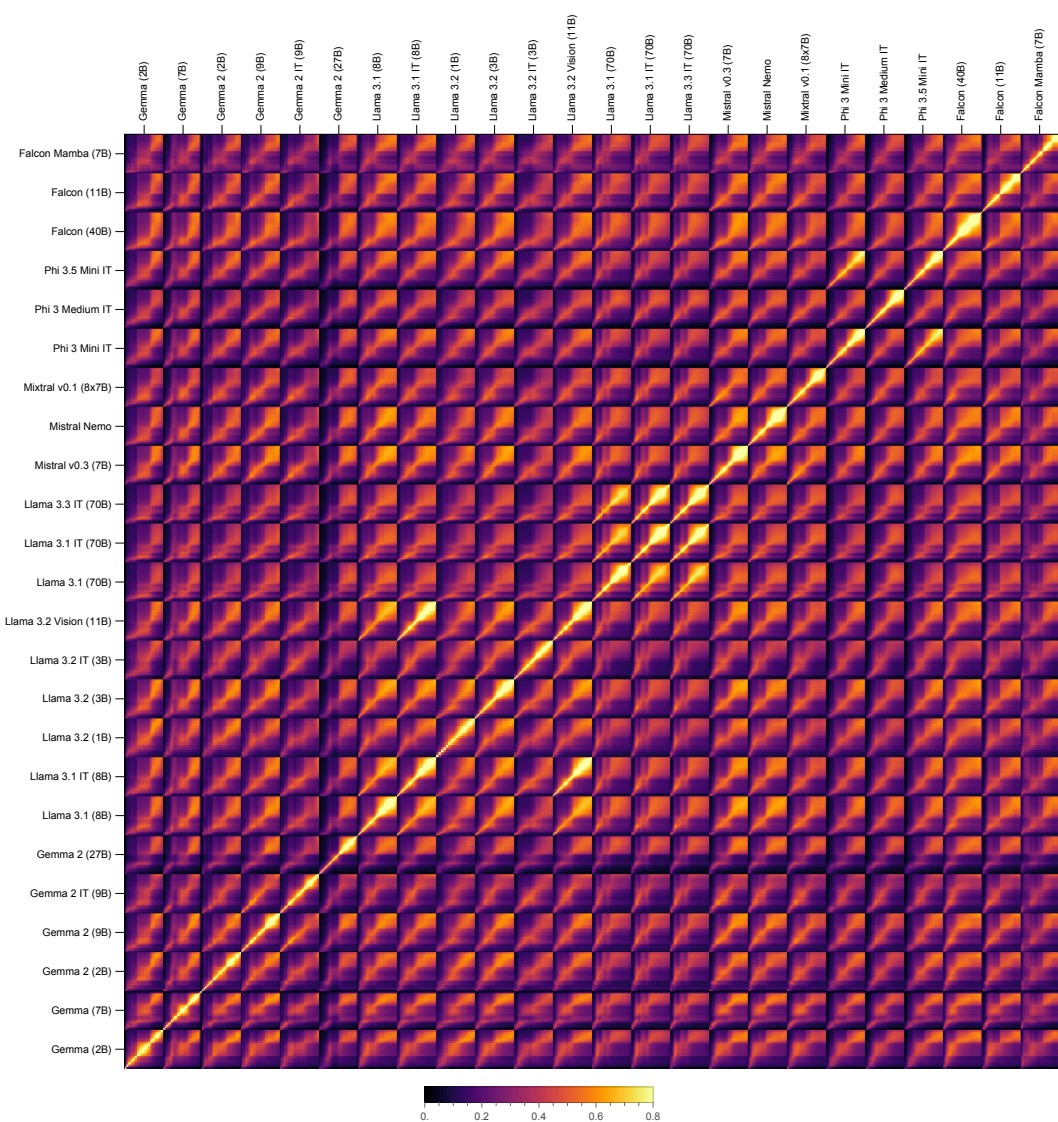

Figure 28: Affinity matrices for embeddings of lead paragraphs from featured articles on Wikipedia.

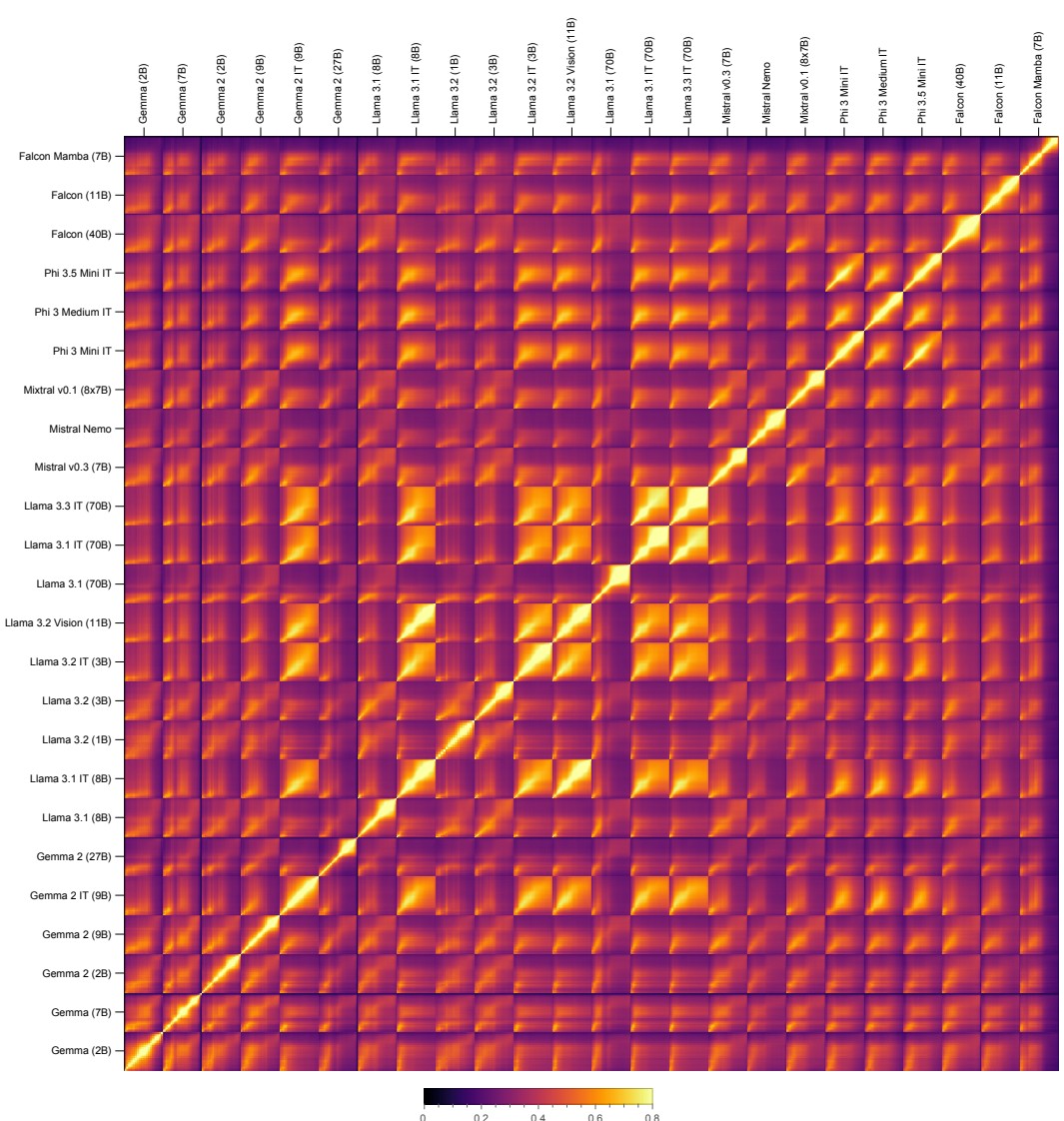

Figure 29: Affinity matrices for embeddings of IFEval (Zhou et al., 2023).

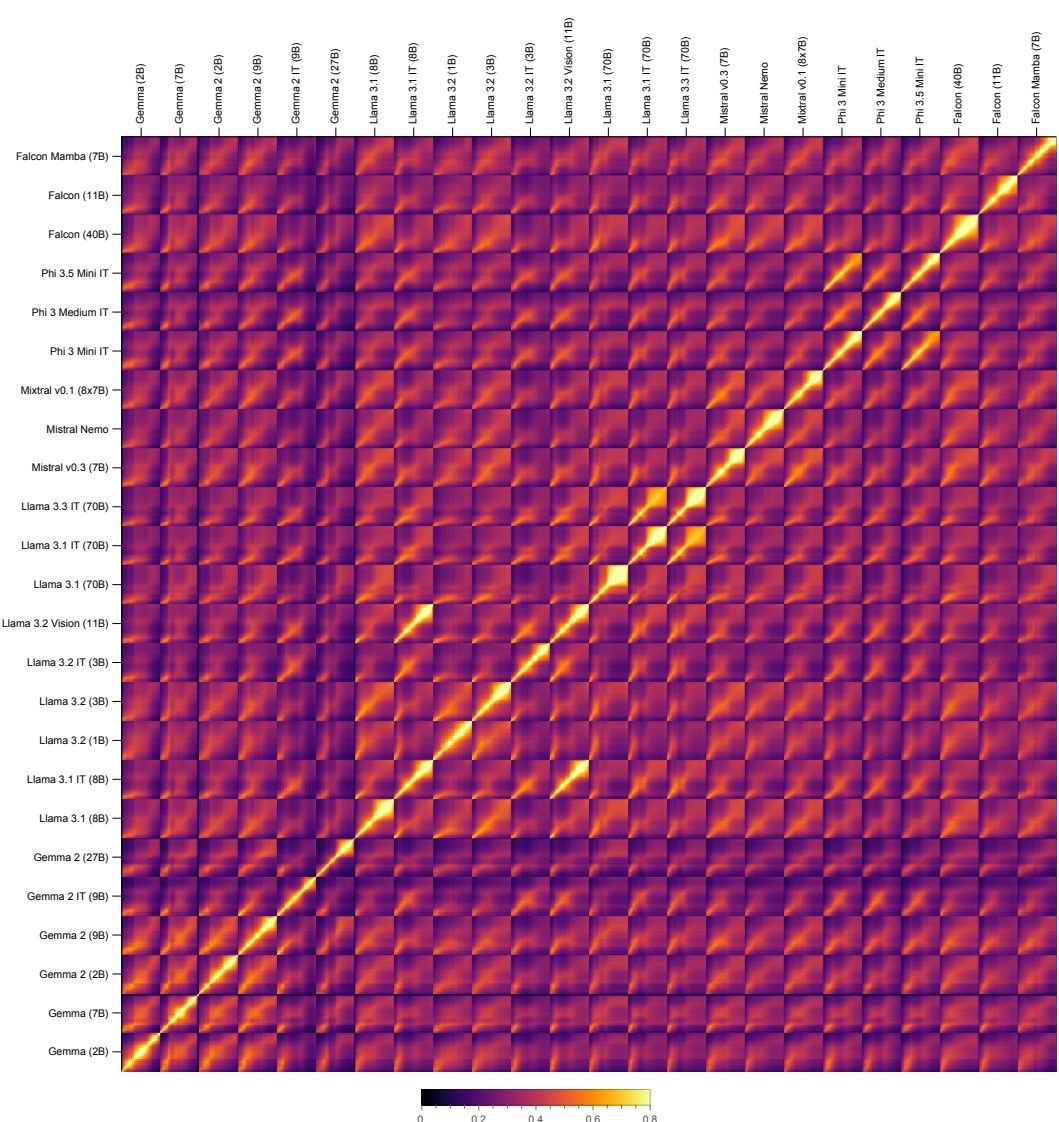

Figure 30: Affinity matrices for embeddings of MMLU (Hendrycks et al., 2021).

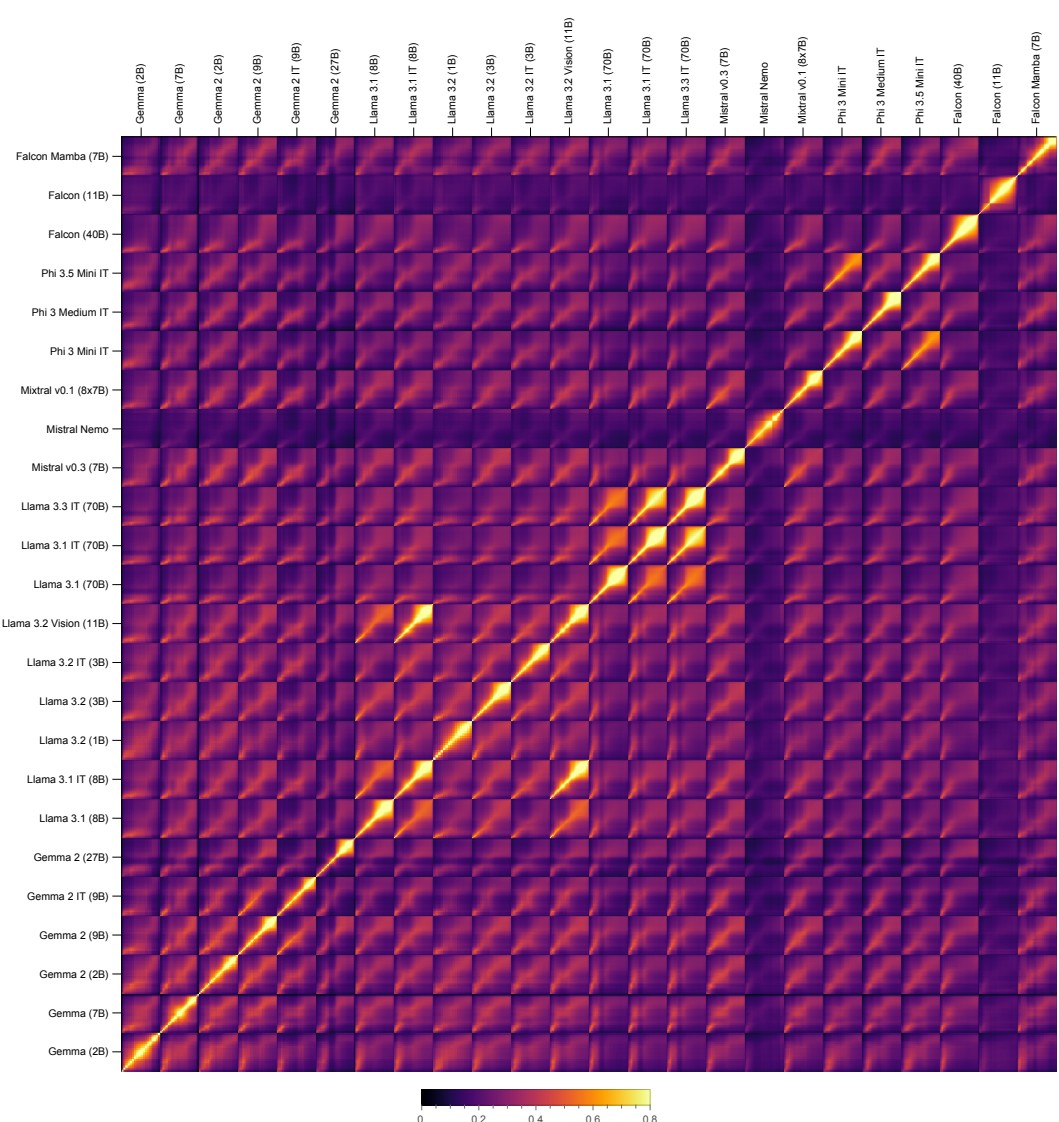

Figure 31: Affinity matrices for embeddings of OPUS Books (English) (Tiedemann, 2012).

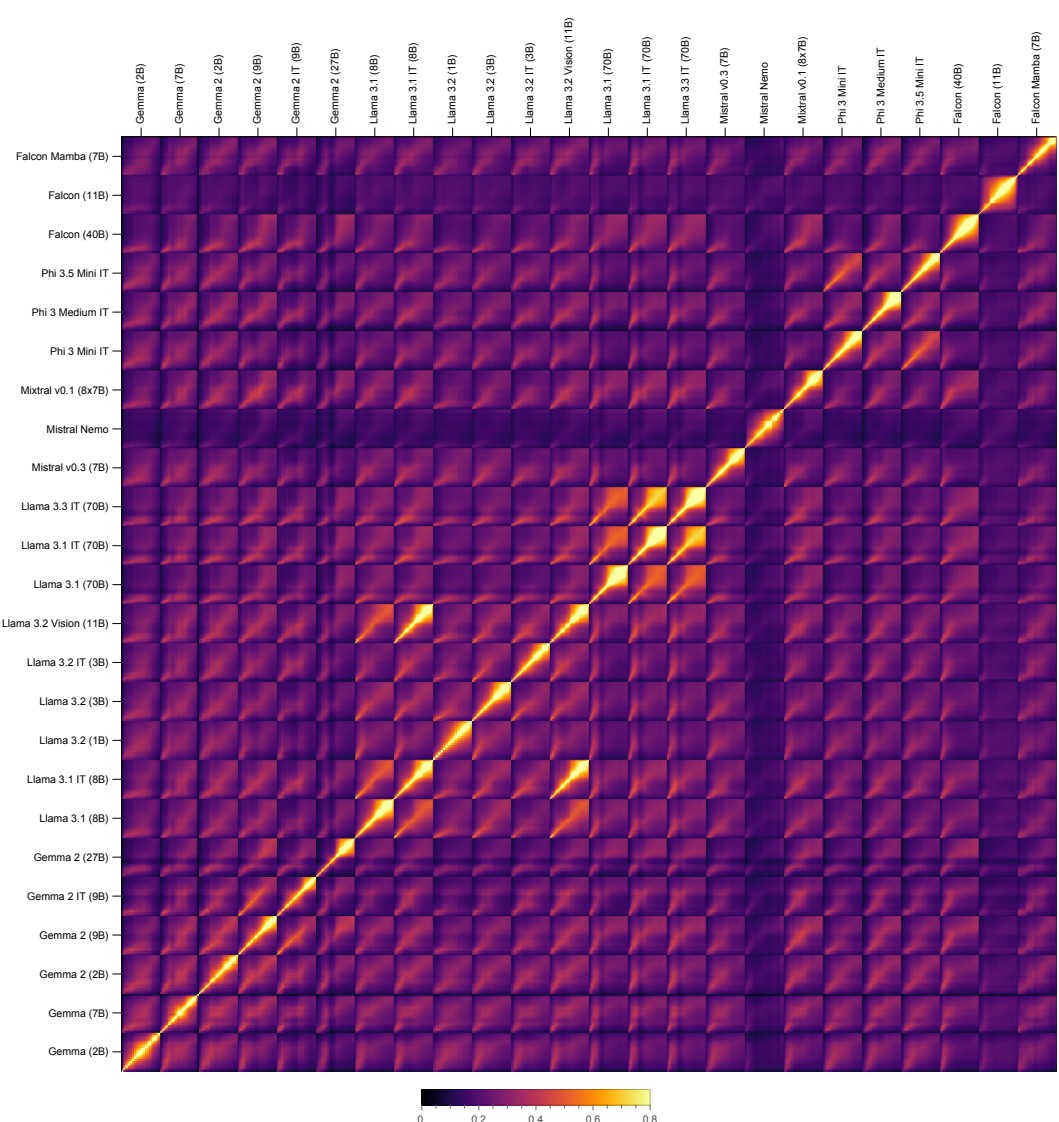

Figure 32: Affinity matrices for embeddings of OPUS Books (German) (Tiedemann, 2012).

