# OpenReview forum: "Layers at Similar Depths Generate Similar Activations Across LLM Architectures"
_colmweb.org/COLM/2025/Conference — COLM 2025_

### Official Review · Reviewer_BgXt · 2025-04-22

**Rating:** 6
**Confidence:** 3
**Ethics Flag:** 1

**Summary:**

This paper makes two main claims: 1) that nearest-neighbor relationships induced by model activations vary highly layer to layer and 2) that these vary less across different models at similar / parallel layers. It shows this by computing the nearest neighbor for each datapoint

By and large, I think that this paper is correct and easy to understand. However, its results (in particular regarding Claim 1) are rather unsurprising, as similar claims have already been discussed at length in existing literature. Much ink has been spilled discussing what LLMs do at each of their layers: detokenization in the lower layers, the beginnings of syntax in the middle layers, and more complex syntax / semantics / retokenization in the upper layers. This discussion has been going on in earnest since at least 4-5 years ago, and I don't think that the consensus has ever shifted away from "models do different things at different layers". This paper carefully notes that it studies representations, not mechanisms, but I'm not sure this is a meaningful distinction in this case.

Claim 2 is somewhat more novel - I am personally unsurprised by it, but I think that less existing work has studied this topic. It is interesting that different models have similar k-nearest-neighbors at similar layers. However, I do think some larger takeaway is missing here: what are the factors underlying the similarities between models? Can we tell which things are being encoded across models? There are techniques - in particular cross-coders - that I think could have answered these more interesting questions within the span of a COLM paper.

Overall, my main concerns for this paper regard novelty - is the content novel, and is there enough novel content? So far, I feel that Claim 1 is not very novel, and Claim 2 could be better explored, given the methods available. While this paper is 9 pages long, the amount of novel content is somewhat smaller than I would expect.

**Reasons To Accept:**

- This paper is well-written and easy to follow. Its methods are simple (a good thing!), and for that reason quite clear.
- In general, I feel that this paper's results are correct. Although they have chosen only one similarity metric (and I would have liked more discussion of / experimentation with alternatives), I personally believe these results are most likely robust to the particular metric chosen.

**Reasons To Reject:**

**Low novelty / Lack of engagement with prior work**: In general, this paper is not very novel; its Claim 1 is one that has already been well-addressed (if indirectly) in past work. This comes in part from a lack of references to the interpretability literature. I think that a thorough read of interpretability review papers, from older ones like [Bertology](https://aclanthology.org/2020.tacl-1.54/), to newer ones like [this one (section 4 in particular)](https://arxiv.org/abs/2405.00208) would be helpful for contextualizing this work within the interpretability literature.
- Claim 1: This flavor of claim has been studied (indirectly) quite a lot. [Tenney et al. (2019)](https://aclanthology.org/P19-1452/) studies what model representations contain at each layer, and finds that each contains different things. While Tenney et al. is a bit outdated at this point, more recent work has spoken to this issue as well. [Lad et al. (2024)](https://arxiv.org/abs/2406.19384?) for example, review some relevant literature; see also work like [this](https://arxiv.org/abs/2401.06102). In general, the review papers I pointed to earlier will show quite a bit of interest in what models do / represent at each layer, and a pretty clear consensus (as I see it) that models have different stages of processing, wherein different (groups of) layers do / represent different (specific) things.
- Claim 2: I think that this claim is somewhat more novel, but could have better been answered in a more satisfying way. Right now, the only conclusion that can be drawn is that models are cross-layer similar according to this one metric / dataset. By expanding this to other metrics / datasets, these results would become a bit more robust. Moreover, other techniques for cross-model comparison, in particular crosscoders ([Lindsey et al., 2024](https://transformer-circuits.pub/2024/crosscoders/index.html)) would provide a little more insight into *what* is driving this similarity. In general, some discussion of SAEs seems appropriate; SAE work has also looked at feature evolution across layers, and found similar features across models. More broadly, follow-ups / ablations / etc. would have strengthened this claim/paper, which right now feels a bit spare.

**Methodological weaknesses**: although I believe that this paper's conclusions are correct, I don't always agree with the experimental setup / interpretation of results.
- The authors always take representations from the last position of an input. This means that their representation comparison is performed only over a limited subset of the dataset - always the last token (it's unclear to me if this is always e.g. a period, which would be quite bad, or just the end of a sentence). There's not a good reason to do this: people often do this in order to extract "sentence representations" from autoregressive LMs, but this is (in my opinion) poor practice. Although the last token of a passage is the only one that can contain information about all prior tokens, autoregressive LMs have no reason to generate a "sentence representation" per se, and such final tokens cannot be treated as one.
- The authors study only 2048 texts. I imagine that this is for computational reasons, but I worry that using such a small number of texts limits the robustness of these results. Results could be somewhat different for other datasets; in general, representational analysis on a small set of data is dangerous; [see this paper](https://arxiv.org/abs/2104.07143) for an example.
- I know that "As such, the particular choice of representational similarity measure is not [y]our focus." but have you experimented with any others to show that your results are robust to this choice? Given the small number of datapoints studied, this would be a rather inexpensive experiment to run.

---

> ### Author Response · Authors · 2025-05-30
>
> Thank you for your feedback and careful reading of our work. We are glad that you appreciated the clarity of our experiments. We agree that it is important for these experiments to be simple and interpretable. We devote the rest of our response to clarifying your questions.
>
> The reviewer is concerned about the **novelty** of Claim 1 in particular. To clarify, we intend for Claims 1 and 2 to be taken together, not individually. Together these claims define the diagonal structure that we observe. As such, we do not argue that Claim 1 is novel on its own, but only that the diagonal structure (as defined by the combination of Claims 1 and 2) is novel. (We will make this more clear in our camera-ready version.)
>
> When we first started this project, we thought---as we suspect the reviewer does---that existing interpretability work must have already shown that (only) layers at similar depths from different models are similar to one another. However, we were surprised to find that despite much peripheral work in interpretability that hints at this, there does not appear to be any existing work that systematically demonstrates this hypothesis empirically. We think our paper is important because it substantiates a claim that is perhaps widely suspected, but which has not been substantiated with systematic evidence. We have also encountered strong but divergent intuition in the literature and among colleagues about whether this phenomenon should hold. Even in this discussion, reviewer j38m was concerned about the validity of Claim 1, while BgXt finds Claim 1 very intuitive. We think this is the value of careful empirical work.
>
> The reviewer also asked about the sensitivity of results to other **datasets**. We perform experiments on a range of datasets which are discussed in-depth in appendix A.
>
> The reviewer also asked about the sensitivity of results to the choice of **similarity measure**. We have performed some experiments with other similarity measures like CKA and found similar results. We focus on the mutual k-NN similarity because measures like CKA are generally less interpretable and can be sensitive to things like the embedding dimension, meaning that one has to be more careful when performing cross-model comparisons. Also, to quote our submission, "We do not claim that the nearest neighbor-based measure we use is the 'correct' way to measure representational similarity (whatever that would mean), but that it shows that there *exist* properties of activations that are shared according to these patterns." In other words, that there exists any similarity measure that can identify pairs of layers from corresponding depths is already significant.
>
> The reviewer suggests that experiments that get at similar questions could be done with **SAEs or crosscoders**. We agree that such experiments could be performed, but they would likely be far more complicated and consequently have more opportunities for misleading results. As the reviewer mentions, part of the strength of our experiments is their simplicity and concreteness. Our method does not require any additional training, alignment, or interpretability assumptions. While methods like SAEs or crosscoders could potentially reveal more interpretable or fine-grained shared features, they introduce additional complexity and possible confounds. That similarity is already present when looking at (arbitrary looking) nearest neighbor relationships is a stronger result than if similarity could be detected using aligned SAEs.
>
> The reviewer also wonders about the potential **causes** of diagonal structure, as well as the content that is similarly represented across models. We agree that this is a natural followup that should be a topic of future research, perhaps using SAEs, or perhaps by simply studying the content of shared nearest neighbor sets. We will add a Future Work section to the camera ready suggesting possible research directions to address these questions.
>
> The reviewer argues that **activations on positions other than the last token** could be studied. We have also run experiments where, instead of looking at the activations on the last token, we looked at the mean of the activations over all the tokens (which is done in some existing work). However, the results were very similar to those we got with last token activations. In general, last token activations can be compared more reliably between models because they are guaranteed to be independent of the tokenization (which can vary from model to model).
>
> The reviewer asks whether the results are sensitive to the **number of input texts** used. We have run experiments with both fewer (1024) and more (4096) examples. The results were very similar to those we got with 2048. We have also run experiments on several disjoint datasets (as mentioned above), which further demonstrates that our results are not an artifact of a particular set of 2048 input texts.

---

> > ### Comment · Reviewer_BgXt · 2025-05-30
> >
> > Thanks for the response! Here are my thoughts:
> >
> > **Novelty**: The fact that the other reviewer finds Claim 1 unlikely makes me appreciate more that some individuals do not consider it settled science (though that reviewer seems to find it so only in light of your results, not the literature more broadly). In general, I'd appreciate it if you engaged more with the literature around Claim 1, including but not limited to the papers that I've cited; if they are not "systematic evidence", you should discuss why. Relatedly, if there is "strong but divergent intuition in the literature", what are the papers whose intuition runs contrary to Claim 1? Given the strong but apparently divergent intuitions to be found therein, more relevant literature could be cited in this submission.
> >
> > **Activation positions**: I am sympathetic to the problems caused by tokenization, but there are ways around this - comparing activations at the end of multi-token words, for example, seems like a reasonable compromise, given the evidence that models detokenize such words at the last token. Averaging activations seems less principled (and prone to issues if activations are of different magnitudes)
> >
> > After this rebuttal, I'm a bit more satisfied with this paper. I think that it could engage better with the literature, because this topic has often been discussed indirectly; still, this direct treatment of the topic could be valuable.

---

### Official Review · Reviewer_VG7E · 2025-05-02

**Rating:** 7
**Confidence:** 4
**Ethics Flag:** 1

**Summary:**

The authors show that the activation latent spaces between different LLMs have a significant overlap, sometimes higher than inter-layer overlap between the same LLMs. To measure this overlap the authors use nearest neighbour analysis of the respresentations over sets of prompts and measure how much they overlap. They observe a diagonal structure that emerges, both when comparing the affinity between layers of the same model but also across models.

**Questions To Authors:**

How does the overlap metric change if examples from multiple datasets are used together? What if the number of examples is different (more than 2048)? Are embeddings at different token positions also similar (eg embedding at the penultimate token position with the last token positon)?

Where is the blue line is figure 6a?

Might be interesting to cite https://arxiv.org/pdf/2503.04429, which found that one can steer models using activations of different models, even though they had to train "translators"

**Reasons To Accept:**

The authors explore these results in a large number of different models and datasets, clarifying a result that had been observed - in multiple different ways - in the literature. Their two claims: that LLMs have different activation geometries at different depths and that they share geometries at similar depths even between different models, seem to be well justified.

**Reasons To Reject:**

It is possible that cosine similarity is not the best metric to measure the similarity between the internal representations, and that the k-nn overlap results don't generalize to more complex metrics of similarity.
The correlation between the activations might not mean that the models use similar representations in similar ways, and no causal experiments were made.

---

> ### Author Response · Authors · 2025-05-30
>
> Thank you for your consideration and careful reading of our work. We are glad that you appreciated our effort in running our experiments on lots of models and datasets. It was one of our core goals to say something not just about some particular pair of models, but about the whole population of modern LLMs, and this took significant effort and compute. We devote the rest of our response to clarifying your questions.
>
> The reviewer asked about use of the **cosine similarity** for comparing embeddings. We chose to use the cosine similarity to measure distance because there are a number of places in transformer architectures where the cosine distance is effectively used. For example, the next token distribution is effectively chosen by the dot product of the last layer activations with the unembedding vectors. As such it is fairly standard practice to use the cosine similarity to compare embeddings.
>
> The reviewer also asked about the sensitivity of results to the choice of **similarity measure**. We have performed some experiments with other similarity measures like CKA and found similar results. We focus on the mutual k-NN similarity because measures like CKA are generally less interpretable and can be sensitive to things like the embedding dimension, meaning that one has to be more careful when performing cross-model comparisons. Also, to quote our submission, "We do not claim that the nearest neighbor-based measure we use is the 'correct' way to measure representational similarity (whatever that would mean), but that it shows that there \emph{exist} properties of activations that are shared according to these patterns." In other words, that there exists any similarity measure that can identify pairs of layers from corresponding depths is already significant.
>
> The reviewer also wonders about the potential **causes** of diagonal structure, as well as the content that is similarly represented across models. We agree that this is a natural followup that should be a topic of future research, perhaps using SAEs, or perhaps by simply studying the content of shared nearest neighbor sets. We will add a Future Work section to the camera ready suggesting possible research directions to address these questions.
>
> To answer the specific questions:
>
> - We have not run experiments where we mixed together multiple datasets. However, we would speculate that nearest neighbors will mostly be within one dataset, so the results would likely be similar (though perhaps scaled by a constant).
>
> - We have run experiments with both fewer (1024) and more (4096) examples. The results were very similar to those we got with 2048.
>
> - We ran experiments where, instead of looking at the activations on the last token, we looked at the mean of the activations over all the tokens (which is done in some existing work). However, the results were very similar to those we got with last token activations. In general, last token activations can be compared more reliably between models because they are guaranteed to be independent of the tokenization (which can vary from model to model).
>
> - The reviewer noted that the caption for figure 6a mentions a blue line that does not appear. This was a typo and we have fixed it.
>
> - Thank you for the reference to the Activation Space Interventions paper. We will have to take a look and see how it relates!

---

> ### Comment · Reviewer_VG7E · 2025-06-01
>
> After reading the authors' responses to mine and other reviewers' questions I'm happy with my assessment and think this article should be accepted.

---

### Official Review · Reviewer_Dzwk · 2025-05-10

**Rating:** 7
**Confidence:** 3
**Ethics Flag:** 1

**Summary:**

This paper is about representational similarity of LLMs at different depths. The paper defines a nearest neighbor structure for every input and compares similarity of nearest neighbors across different models at different depths.

The main finding is that there is a diagonal structure, i.e., roughly, models have similarly nearest neighbors at similar depths.

**Questions To Authors:**

- Why is Fig. 5 not discussed in the text?

- In Def. 3, what is x? Should it be t? There is also no t in this definition, at odds with line 164. (In general, please check your math formula, as NLP reviewers and authors often do not)

**Reasons To Accept:**

- Well-written paper
- on an interesting research question
- with really interesting results

Overall, I think, I find this paper strong enough to appear at the main conference, because of its interesting results, even despite some weaknesses, as discussed below.

**Reasons To Reject:**

- There is no clear explanation why different models may behave similar at similar depths

- I would find that some more hard numbers would be helpful, i.e., how much more similar are diagonal elements than non-diagonal entries, besides the visual impression you give in the figures and the significance tests in Section 4.1.

---

> ### Author Response · Authors · 2025-05-30
>
> Thank you for feedback and careful consideration of our paper. We agree that highlighting the phenomenon of diagonal structure and substantiating it empirically is a central contribution and grateful that you appreciated it. We will use the rest of this response to clarify your other questions.
>
> The reviewer is concerned that although diagonal structure is visible in affinity matrices, they would like to see more numerical reporting of its **absolute magnitude**. This is a great point. As the absolute similarity at corresponding layers is relatively high (see figures 5b and 6b), we should perhaps explore if there are ways that we could visualize this in a way that makes the numerical scale clearer. At the same time, our emphasis is on the patterns of similarity more so than their absolute values. That the most similar layers are at corresponding depths is significant regardless of the absolute similarity (we have some discussion of this in appendix A.3).
>
> The reviewer also wonders about the potential **causes** of diagonal structure. We agree that this is a natural followup that should be a topic of future research, perhaps using SAEs, or perhaps by simply studying the content of shared nearest neighbor sets. We will add a Future Work section to the camera ready suggesting possible research directions to address these questions.
>
> To answer the specific questions:
>
> - The reviewer asks whether figure 5 is mentioned in the main body. I believe the subfigures of figure 5 are referenced in the main text on lines 227 and 249.
>
> - The reviewer correctly notes that $t$ is supposed to say $x$ in definition 3. We will make that revision.

---

> > ### Comment · Reviewer_Dzwk · 2025-06-05
> >
> > Thanks for the response.
> >
> > Regarding the numerical numbers, I believe that some non-diagonal elements are also quite high. For example, in Fig. 5(c) the diagonal is quite "large", i.e., also several off-diagonal elements are large in magnitude.
> >
> > That said, I have no further comments and will keep my score (no brilliant paper, but worthy to accept, I assume).

---

### Official Review · Reviewer_j38m · 2025-05-12

**Rating:** 7
**Confidence:** 4
**Ethics Flag:** 1

**Summary:**

This paper study the nearest neighbor relationships in hidden layers across different LLM models. The paper makes two observations: 1. Activations collected at different depths within the same model tend to have different nearest neighbor relationships. 2. Activations collected at corresponding depths of different models tend to have similar nearest neighbor relationships. Importantly, the similarity respects the relative depths (e.g., lower layers are similar to lower layers) suggesting representations evolve in similar pattern across LLMs.

**Questions To Authors:**

I have some suggestions:

1. Decay Rate, as mentioned earlier, we can see consecutive layers in one model have high affinity. We can measure the relationship between affinity and distance (maybe normalize total depth) in layers (within the same model and across models）. For cross models study, maybe we need to pick the maximum affinity layer first, then do the decay rate computation.

2. I would put the diagonal structure （within the same model and across models）as the main message instead of claim 1 and 2

I really like this paper and its' finding, but feels the presentation needs more work.

**Reasons To Accept:**

1. The diagonal structure is an interesting finding.
2. Substantial experimental efforts of many models and queries.

**Reasons To Reject:**

1. Claim 1 and 2 are a bit vague, but given claim 2 being true, one can argue claim 1 might be false [if you are happy with the affinity across models, you should probabily be happy with the affinity of two random layers within the same model]. In appendix E1, one can see that the big diagonal shines the most, and **consecutive layers** in one model have high affinity. In paritucular, different layers in Llama models appear to have higher affinity than their affinity against a similar layer in another model.  Clearly, we should get more detailed numerical results to support claim 1.

---

> ### Author Response · Authors · 2025-05-30
>
> Thank you for feedback and careful consideration of our paper. We are glad that you appreciated our effort in running experiments on a wide range of models and datasets. It was important to us to have experiments that represented modern LLMs generally and not just a particular pair of models. We will use the rest of this response to clarify your other questions.'
>
> The reviewer suggests centering **diagonal structure as the main message** (rather than Claims 1 and 2). We agree; however, to clarify, we intend for Claims 1 and 2 to be *taken together as defining* the diagonal structure that we observe. This is what we meant when we wrote that the claims "seem unremarkable on their own, but together are quite surprising"). We will clarify this point in the camera-ready version.
>
> The reviewer also notes that consecutive layers within the same model appear to show high similarity and asks how this relates to **Claim 1**. We agree that consecutive layers within a single model often generate similar activations (which would be expected because of residual connections). However, more distant layers tend to generate increasingly different activations. This can be seen indirectly even in cross-model affinity matrices: If all the layers in a model generated similar representations, then you would not see diagonal structure in cross-model comparisons because you would have that $A_{i,j} \approx A_{k,j}$ which would not give diagonal structure. Diagonal structure in cross-model comparisons is only possible because layers within a model can be distinguished by depth (Claim 1).
>
> We also agree that more **numerical description** of diagonal structure generally would be helpful, and have already prepared expanded statistical tests of diagonal structure which we plan to include in the camera ready.
>
> Finally, the reviewer suggests studying how similarity **"decays"** as layers get further apart. We actually generated decay curve plots that show similarity as a function of the distance between layers, but did not end up including them. In particular, we never found a good way for visualizing these decay curves for cross-model comparisons. However, the reviewer makes a good case that even within-model decay curve plots would be valuable addition which we will include in the camera ready.

---

> > ### Comment · Reviewer_j38m · 2025-06-05
> >
> > At least, the diagonal structure should be mentioned in the abstract.
> >
> >
> > Sinces other reviewers all like the finding, and we do believe the presentation can be improved. I will raise my evaluation. Sorry for the bit late reply.
> >
> > btw "numerical description of diagonal structure " , it would be nice to include them in the rebuttal here.

---

### Decision · Program_Chairs · 2025-07-08

**Decision:**

Accept

**Comment:**

Reviewers overall liked the paper. Authors properly responded to comments. Perhaps reorient some of the claims around the diagonal structure or emphasise it a bit more.